# CHD7 regulates otic lineage specification and hair cell differentiation in human inner ear organoids

Jing Nie [1], Yoshitomo Ueda [1], Alexander J. Solivais[1] & Eri Hashino [1,2] ✉

Mutations in *CHD7* cause CHARGE syndrome, affecting multiple organs including the inner ear in humans. We investigate how *CHD7* mutations affect inner ear development using human pluripotent stem cell-derived organoids as a model system. We find that loss of CHD7 or its chromatin remodeling activity leads to complete absence of hair cells and supporting cells, which can be explained by dysregulation of key otic development-associated genes in mutant otic progenitors. Further analysis of the mutant otic progenitors suggests that CHD7 can regulate otic genes through a chromatin remodeling-independent mechanism. Results from transcriptome profiling of hair cells reveal disruption of deafness gene expression as a potential underlying mechanism of CHARGE-associated sensorineural hearing loss. Notably, co-differentiating *CHD7* knockout and wild-type cells in chimeric organoids partially rescues mutant phenotypes by restoring otherwise severely dysregulated otic genes. Taken together, our results suggest that *CHD7* plays a critical role in regulating human otic lineage specification and hair cell differentiation.

CHARGE syndrome is a congenital multi-organ disorder mainly caused by de novo mutations in the *CHD7* gene[1], which encodes an ATP-dependent chromatin remodeling protein that regulates target genes expression via changes in nucleosome accessibility[2]. The most prevalent clinical features of CHARGE syndrome include malformation of the inner ear structures, accompanied by pre-lingual deafness and vestibular dysfunctions[3–5]. Studies in *Chd7* mutant mouse models demonstrated morphological defects in the semicircular canals and the cochlea, which is accompanied by head-bobbing and circling behaviors consistent with vestibular dysfunctions[6–11]. In heterozygous *Chd7* mutants and in mice with *Atoh1*-driven *Chd7* conditional knockout, hair cells develop normally and display normal stereocilia morphology[8]. However, hair cells in the conditional knockout were subsequently degenerated after the onset of hearing due to hypersensitivity to stress, leading to profound sensorineural hearing loss[12]. These and other mouse model studies provided important insights into CHARGE syndrome-associated inner ear phenotypes[13–16]. However, despite a small

number of *Chd7* downstream genes, including *Sox2*, *Hmx3*, and *Jag1*, as well as an upstream regulator *let-7* being identified previously[10,17], it remains unclear how the loss of *CHD7* affects gene expression at the transcriptome level in key otic lineage cell types such as otic progenitors, hair cells, and supporting cells. We have previously established a pluripotent stem cell-derived inner ear organoid system capable of recapitulating inner ear development through sequential generation of non-neural ectoderm (NNE), otic-epibranchial progenitor domain (OEPD), otic placodes/pits, and otic vesicles. The otic progenitor cells in otic vesicles undergo self-guided differentiation and form mechanosensitive hair cells, supporting cells, and neurons forming contacts with the hair cells[18–21].

In this study, we generate multiple *CHD7* mutant human embryonic stem cell (hESC) lines, including a complete knockout (KO) and a patient-specific missense mutant. By modeling CHARGE syndrome in inner ear organoids derived from these cell lines, we demonstrate that CHD7 plays critical roles in otic lineage specification and hair cell generation.

[1]Department of Otolaryngology-Head and Neck Surgery, Indiana University School of Medicine, Indianapolis, IN 46202, USA. [2]Stark Neurosciences Research Institute, Indiana University School of Medicine, Indianapolis, IN 46202, USA. ✉e-mail: ehashino@iu.edu

## Results

### CHD7 is expressed throughout key otic developmental stages

We first accessed spatiotemporal changes in *CHD7* expression during human inner ear organoid differentiation. To circumvent the low specificity of available CHD7 antibodies (Supplementary Fig. 1), we tagged the endogenous *CHD7* gene with 3×Flag in hESCs with CRISPR (Supplementary Fig. 2). Anti-Flag detection of CHD7-3×Flag eliminated nearly all non-specific bindings (Fig. 1a) and showed *CHD7* expression in all early otic developmental stages, including NNE, OEPD, otic pits, and otic vesicles (Fig. 1b–f). In addition, *CHD7* is strongly expressed in hair cells and, at lower levels, in supporting cells (Fig. 1g). These results reveal that *CHD7* is expressed in key otic-lineage cell types throughout all stages of inner ear development.

### Loss of CHD7 causes failure in sensory epithelium formation

To recapitulate the phenotypic manifestation of CHARGE syndrome in inner ear organoids, we next created mono- and bi-allelic *CHD7* KO hESC lines by targeting the first of the 38 coding exons of *CHD7* with CRISPR. Frameshift indel formation at this early coding region leads to nonsense-mediated mRNA decay (NMD), which eliminates *CHD7* transcripts[22]. If the mutated transcript escapes the NMD mechanism, it results in early truncation of the CHD7 protein prior to any functional domain, therefore creating a null deletion. We chose two clonal lines with frameshift indels in one, or both alleles, and designated them as *CHD7*[KO/+] and *CHD7*[KO/KO] hESC lines for further analysis (Fig. 2a and Supplementary Figs. 3–4). Western blotting confirmed the complete elimination of CHD7 protein in *CHD7*[KO/KO] mutant, and reduced protein expression in *CHD7*[KO/+] (Fig. 2b). When differentiated into inner ear organoids, both mutant lines generated morphologically normal PAX2+ PAX8+ EPCAM+ otic vesicles at differentiation day 20 (d20) (Fig. 2l, t). Consistent with the normal morphology of stereocilia-bearing hair cells in heterozygous *Chd7* deficient mice[8], at day 70 (d70), the *CHD7*[KO/+] mutant organoids generated hair cells and supporting cells that are indistinguishable from the WT control. In addition, these mono-allelic KO hair cells exhibited stereocilia with normal morphology (Fig. 2m–o). In contrast, neither hair cells nor supporting cells were observed in *CHD7*[KO/KO] organoids (Figs. 2u–w and 7, Supplementary Figs. 15–17) (n = 233 aggregates from 7 independent organoid cultures).

In addition to the complete knockouts, we also introduced a patient-specific missense mutation in *CHD7* in hESCs. A heterozygous serine to phenylalanine substitution at the CHD7 residue 834 (p.S834F, c.2501 C > T) was described by two independent clinical studies in three CHARGE patients and in a patient with idiopathic hypogonadotropic hypogonadism[23,24]. This missense mutation occurs in a highly conserved sequence motif at one of the chromodomains of CHD7. Previous biochemical analysis demonstrated that this single amino acid substitution completely abolishes CHD7's ATPase activity. Consistent with this mutant protein's inability to hydrolyze ATP, its chromatin remodeling activity is also completely abolished[2]. We used a CRISPR base editor[25] and created mono- and bi-allelic *CHD7* S834F mutations in hESCs (Fig. 2c and Supplementary Fig. 5). When differentiated towards the otic lineage in inner ear organoids, the mono-allelic and bi-allelic *CHD7* S834F mutants essentially phenocopied the corresponding KO phenotypes at both d20 and d70. The *CHD7*[S834F/+] mutant gave rise to morphologically normal otic vesicles, supporting cells, and stereocilia-bearing hair cells (Fig. 2h–k), while no hair cells or supporting cells were found in the *CHD7*[S834F/S834F] mutant organoids (n = 210 aggregates from 5 independent organoid cultures) despite the presence of normal-looking otic vesicles (Fig. 2p–s). Collectively, these results demonstrate that hair cell and supporting cell derivation require the ATP-dependent chromatin remodeling activities of CHD7.

### Loss of CHD7 causes dysregulation of otic development genes

To investigate the mechanisms underlying the failure in hair cell and supporting cell generation in d70 *CHD7*[KO/KO] organoids, we performed transcriptome profiling in the seemingly normal d20 *CHD7*[KO/KO] otic progenitor cells using scRNA-seq. We previously generated a *PAX2*−2a-nGFP (*PAX2*[nG]) reporter hESC line to label the otic progenitor cells with nuclear GFP[26], and all of our four *CHD7* mutant lines were built on this *PAX2*[nG] genetic background (Supplementary Fig. 6). To enrich otic progenitors, we dissociated d20 WT and *CHD7*[KO/KO] organoids and FACS-isolated *PAX2*[nG+] cells (Supplementary Fig. 7a). scRNA-seq analysis revealed that 89.6% of WT *PAX2*[nG+] cells are otic progenitors. As neuroblast cells that delaminate from the otic vesicles and hindbrain neurons are also known to be *PAX2*-positive[27], these cell populations were also present in the WT dataset. In *CHD7*[KO/KO] samples, otic

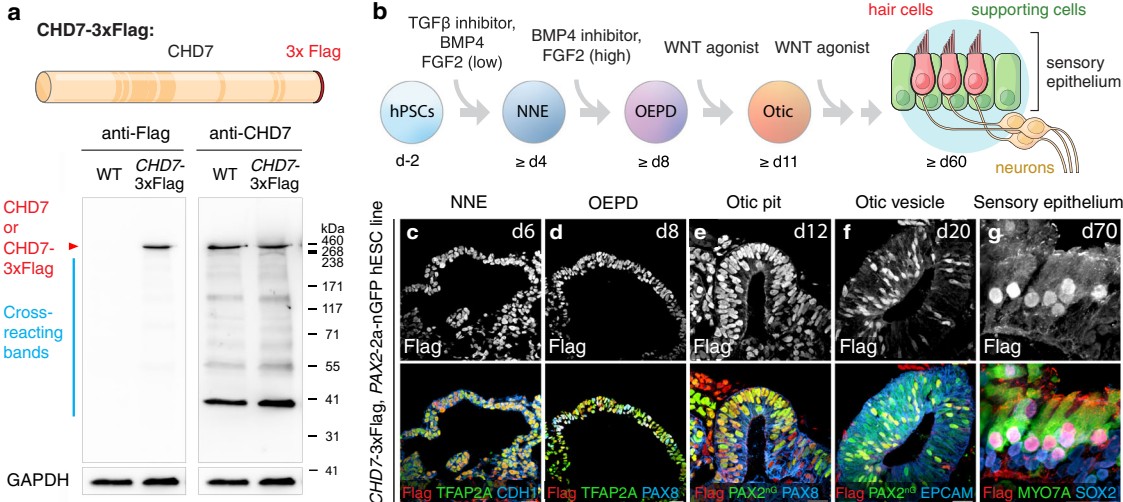

**Fig. 1 | CHD7 is expressed at key otic development stages. a** Western blotting of WT (*PAX2*−2a-nGFP cell line, hereafter *PAX2*[nG]) and *CHD7*−3×Flag hESCs using anti-Flag and anti-CHD7 antibodies. Calculated molecular weight of CHD7 and CHD7-3×Flag are 336 kDa and 339 kDa, respectively. Source data are provided as a Source Data file. **b** Schematics of otic lineage differentiation during human inner ear organoid culture. Schematics adapted from Nie J, Hashino E. (2020) Generation of inner ear organoids from human pluripotent stem cells. Methods in Cell Biology,

159: 303–321, with permission from Elsevier. **c–g** Immunostaining at key otic development stages in *CHD7*−3×Flag *PAX2*[nG] human inner ear organoids using an anti-Flag antibody, as well as antibodies against NNE markers TFAP2A and CDH1, OEPD markers TFAP2A and PAX8, otic placode/pit and otic vesicle markers PAX2[nG], PAX8, and EPCAM, and hair cell markers MYO7A and SOX2 and supporting cell marker SOX2. Scale bars, 25 μm.

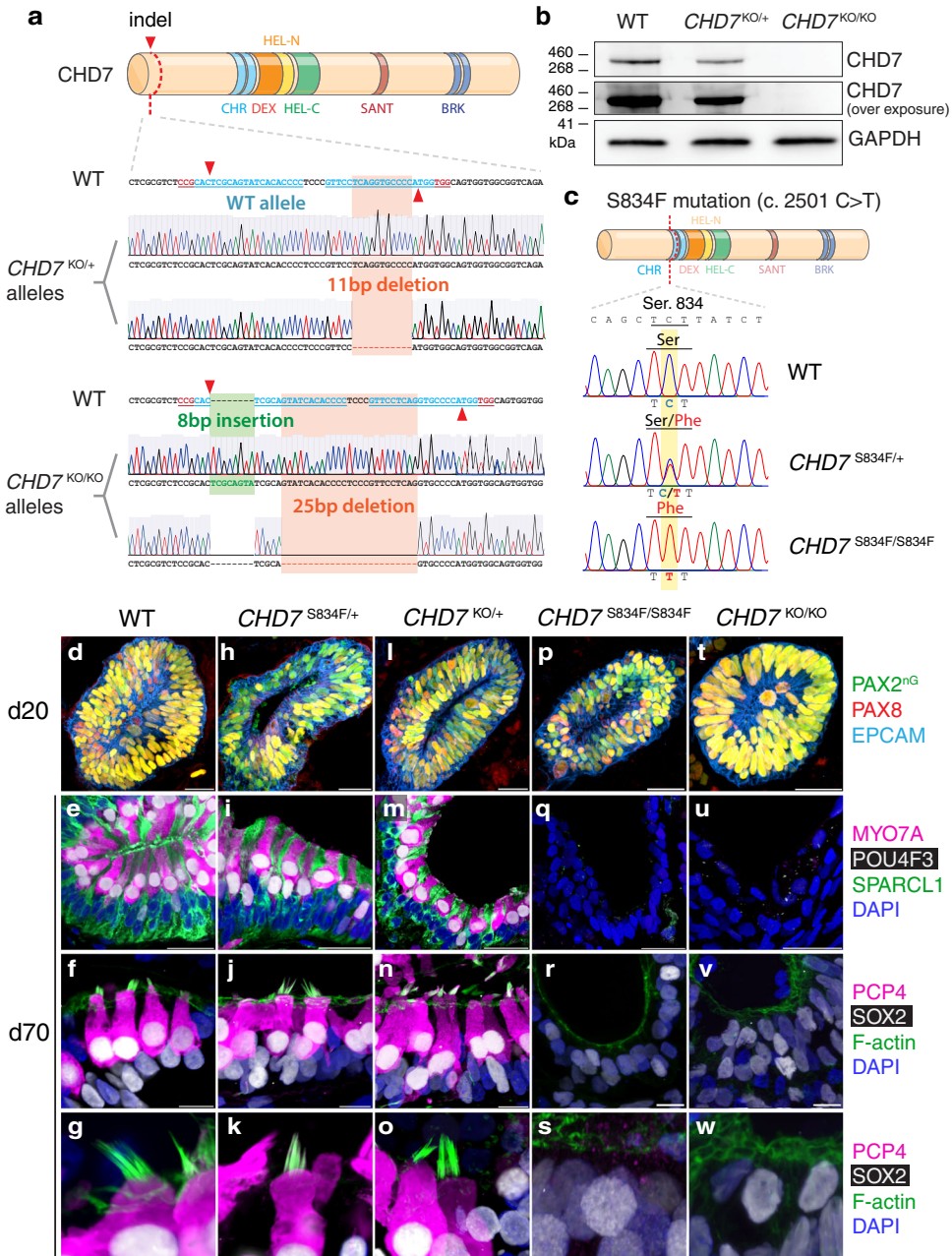

**Fig. 2 | CHD7 and its ATP-dependent chromatin remodeling activities are required for sensory epithelium derivation. a** Sanger sequencing chromatograms of *CHD7*[KO/+] and *CHD7*[KO/KO] alleles cloned into TOPO vectors. **b** Western blotting of WT (*PAX2*[nG]), *CHD7*[KO/+], and *CHD7*[KO/KO] hESCs using an anti-CHD7 antibody. Source data are provided as a Source Data file. **c** Sanger sequencing chromatograms of WT (*PAX2*[nG]), *CHD7*[S834F/+], and *CHD7*[S834F/S834F] hESCs at the *CHD7* c.2501 locus. **d–w** Immunostaining of d20 and d70 WT and *CHD7* mutant organoids. Antibodies highlight otic progenitors (PAX2[nG], PAX8, and EPCAM), hair cells (MYO7A, POU4F3, PCP4, SOX2, and F-actin for stereocilia of hair cells), and supporting cells (SPARCL1 and SOX2). Scale bars, 25 μm (top two rows of **d**–**w**), 10 μm (third row of **d**–**w**), and 5 μm (bottom row of **d**–**w**).

progenitors made up a smaller percentage (39.7%), while two additional clusters of *PAX2*[nG+] *EPCAM*[−] non-epithelial cells made up more than half of all mutant cells, with one of these two clusters showing high levels of cell cycle marker gene expression. In addition to these abnormal *PAX2*[+] non-epithelial cells, *CHD7* depletion also appeared to affect the neuroblast cells, as the mutant cells only made up 4.9% of all neuroblast cells (Fig. 3a–b and Supplementary Fig. 7b, d).

Since it is the otic progenitor cell population that gives rise to hair cells and supporting cells, we focused on otic progenitors for further analysis (Fig. 3b and Supplementary Fig. 7c). Differential gene expression analysis revealed 323 upregulated genes and 129 downregulated genes in *CHD7*[KO/KO] otic progenitors relative to the WT

control (fold change ≥ 2.0, $P \le 1 \times 10^{-10}$) (Fig. 4a). Among these differentially expressed (DE) genes, 15 deafness genes listed in the OtoSCOPE gene panel[28] were downregulated, including *TBX1*, *LMX1A*, and *SOX10* (Fig. 3c). Gene set enrichment (GSE) analysis of the downregulated genes using the iDEA pipeline[29] suggested that genes in many inner ear development-related Gene Ontology (GO) categories were disrupted, including *DLX5* and *SIX1* in the inner ear morphogenesis GO term (Figs. 3c, 4d). In addition, GSE analysis also revealed dysregulation of FGF and WNT signaling pathways, as well as enrichment of multiple gene sets closely related to the cellular functions of the otic progenitors, such as cell junction organization and extracellular matrix organization (Fig. 4d). To examine whether cell lineage

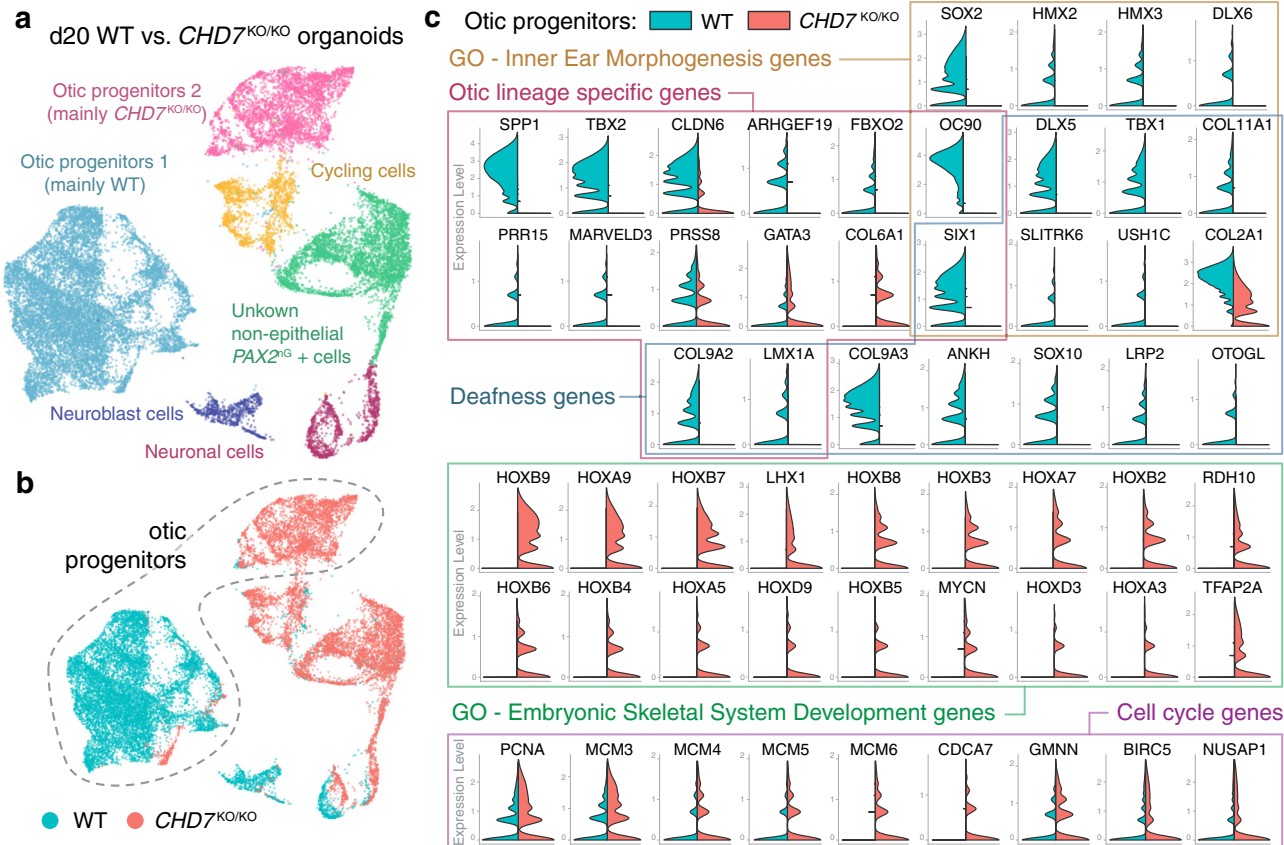

**Fig. 3 | scRNA-seq of d20 WT and *CHD7*^KO/KO^ organoids revealed critical role of CHD7 in otic development. a, b** Uniform manifold approximation and projection (UMAP) plot and cluster annotation of FACS-isolated *PAX2*^nG^-positive cells from d20 WT and *CHD7*^KO/KO^ organoids as grouped by cluster (**a**) or by genotype (**b**). Dotted line highlights the two otic progenitor clusters. Data represent 22,390 cells. **c** Split-violin plots of key genes dysregulated in *CHD7*^KO/KO^ otic progenitors compared to the WT otic progenitors.

identity is affected, we compared our dataset with a list of otic lineage-specific genes systematically identified by Hartman et al.[30]. While 57.7% of these otic-specific genes remained largely unaffected, 38.5% of them, including the highest ranked otic-specific genes *FBXO2*, *COL9A2*, and *OC90*, were significantly downregulated in *CHD7*^KO/KO^ otic progenitors, suggesting that the otic identity is partially impaired (Figs. 3c and 4b). Consistent with the aberrant otic gene expression profile, a large number of genes not normally found in the developing otic vesicle were significantly upregulated, including a cohort of HOX genes (e.g., *HOXB9*, *HOXA7*, and *HOXD3*) from the embryonic skeletal system development GO category (Fig. 3c). We also noticed upregulation of multiple cell cycle marker genes (e.g., *PCNA* and *MCM3*) (Fig. 3c), which could explain the extensive expansion of *PAX2*^+^ epithelial vesicle structures in -d25 *CHD7*^KO/KO^ organoids (Fig. 4c and Supplementary Fig. 6). Taken together, these results suggest that the failure of hair cell and supporting cell derivation from the *CHD7* null mutant stemmed from a multitude of dysregulation at the gene and gene set levels, including downregulation of genes essential to hearing and inner ear development, dysregulation of components and regulators of signaling pathways, cell junction, and extracellular matrix, dysregulation of cell cycle control, as well as, a partially drifted otic lineage identity.

**Otic genes regulated independently of chromatin remodeling**
To determine differential gene expression in the *CHD7*^S834F/+^, *CHD7*^KO/+^, and *CHD7*^S834F/S834F^ mutants, as well as to confirm the *CHD7*^KO/KO^ scRNA-seq results, we next examined protein expression of 7 key markers with all four *CHD7* mutant lines. While SOX2 expression followed gene-dosage changes of *CHD7* and showed moderate dysregulation in the

mono-allelic mutants and more severe dysregulation in the bi-allelic mutants (Fig. 5a–g), there were markers not following this trend. For example, COL9A2 was downregulated in the bi-allelic mutants to the same extent as the mono-allelic mutants, while the non-otic HOXB9 protein was not upregulated in the two mono-allelic mutants, indicating a more faithful otic lineage identity. In addition, SOX10 showed an opposite direction of dysregulation between the mono- and bi-allelic mutants (Fig. 5v–ab, aj–aw). These results suggest a complex dysregulation pattern of *CHD7* downstream proteins, which did not simply follow the changes in *CHD7* gene dosage.

Regarding S834F and its corresponding mono- or bi-allelic KO mutant, while most of them showed comparable levels of dysregulation, there are several exceptions. For example, SIX1 and FBXO2 showed significant downregulation in *CHD7*^KO/KO^ compared to *CHD7*^S834F/S834F^, and DLX5 showed significant downregulation in *CHD7*^KO/+^ compared to *CHD7*^S834F/+^ (Fig. 5h–u, ac–ai). Considering the complete abolishment of ATPase and chromatin remodeling activities in the S834F mutant[2], the differential downstream protein expression between S834F and its corresponding KO mutant reveal the presence of CHD7 function(s) beyond its ATPase and chromatin remodeling activities. Such chromatin remodeling-independent mechanism appears to be solely responsible for regulating key otic genes such as *FBXO2* (Fig. 5ag–ai).

When examining SOX2 expression, we noticed the presence of SOX2^+^ cells outside of PAX2^nG+^ EPCAM^+^ otic vesicles (Fig. 5b–f). To determine their identity as well as to investigate the identities of other PAX2^nG−^ populations, we performed scRNA-seq analysis on both the *PAX2*^nG+^ and *PAX2*^nG−^ populations of d20 WT organoids. Marker gene profiling suggests that the d20 *PAX2*^nG−^ populations are composed of

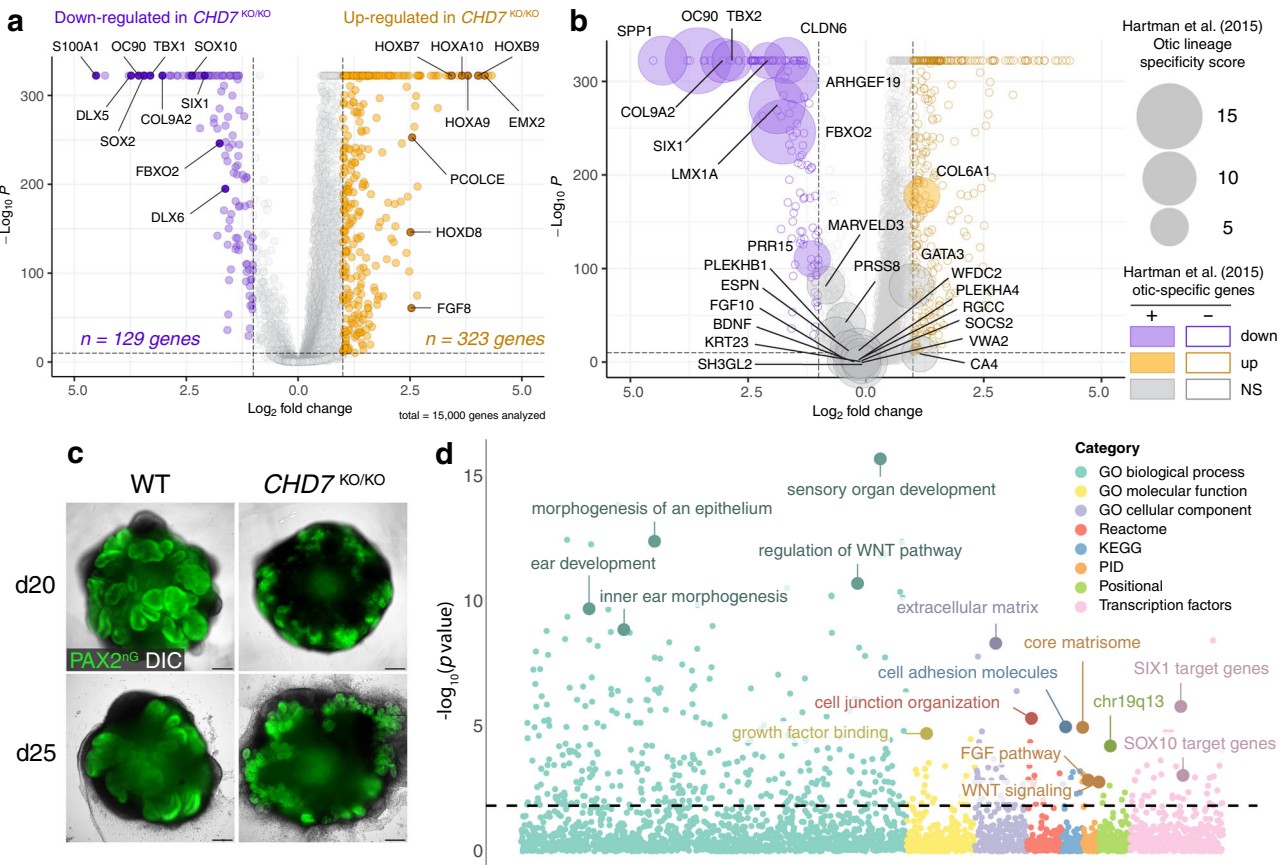

**Fig. 4 | scRNA-seq of d20 WT and *CHD7*KO/KO organoids suggest CHD7 plays essential role in otic development. a** Volcano plot of differentially expressed genes in *CHD7*KO/KO otic progenitors. **b** Bubble plot of dysregulated and unaffected otic lineage-specific genes in *CHD7*KO/KO otic progenitors. The areas of the bubbles represent the gene expression fold-change values of E10.5 otic vesicles versus non-otic tissues as reported by Hartman et al. (2015) **c** Live imaging of d20 and d25 WT and *CHD7*KO/KO organoids. **d** Bubble plot of enriched gene sets from downregulated genes in d20 *CHD7*KO/KO otic progenitors. For differential expression (DE) analysis in (**a** and **b**), the two-sided test DESeq2 was used, and adjustments were made by the Benjamini-Hochberg method for multiple comparisons. For gene set enrichment analysis (GSEA) using the one-sided test Integrative Differential expression and gene set Enrichment Analysis (iDEA) shown in (**d**), adjustments of *p*-values were made by the Louis method. Scale bars, 250 μm.

neuroblast cells, neurons, neural crest cells, and mesenchymal cells (Supplementary Fig. 8). The SOX2+ cells outside of otic vesicles in d20 WT and *CHD7*KO/KO organoids (Fig. 5b, f) appear to be S100B+ neural crest cells (Supplementary Figs. 8f and 9a–f).

**Deafness genes are dysregulated in *CHD7*KO/+ hair cells**

To investigate how decreased *CHD7* expression affects the morphologically normal hair cells and supporting cells at the transcriptome level, we performed scRNA-seq with d70 WT and *CHD7*KO/+ organoids. We first labeled the hair cells with a highly specific *POU4F3*–2a-ntdTomato (*POU4F3*nT) fluorescence reporter in the WT and the *CHD7*KO/+ genetic backgrounds (Supplementary Fig. 10)[26]. We FACS-separated the *POU4F3*nT+ and *POU4F3*nT- cells from micro-dissected WT and *CHD7*KO/+ d70 organoids (Supplementary Fig. 11a) and performed scRNA-seq of these four groups of cells in four separate reactions. As hair cells only constitute ~1–2% of all cells in inner ear organoids[31], this experimental design allowed an adequate number of hair cells to be collected for downstream analysis. Indeed, we obtained 9,884 hair cells (28.5%) and 6,273 supporting cells (18.1%) when analyzing the merged dataset (Fig. 6a and Supplementary Fig. 11b–e). Unsupervised cell clustering grouped hair cells into three clusters; one of them had immature hair cell gene expression profiles. The mature hair cells segregated into WT and *CHD7*KO/+ clusters, while the WT and *CHD7*KO/+ supporting cells were intermingled in one cluster (Fig. 6b and Supplementary Fig. 11f–g), which is consistent with the lower expression

levels of *CHD7* in supporting cells and therefore a lesser extent of influence (Fig. 1g and Supplementary Fig. 18a–b). In contrast to the upregulation-oriented differential expression pattern in d20 otic progenitors (Fig. 4a), the majority of DE genes in d70 hair cells and supporting cells were downregulated (fold change ≥ 2.0, $P \leq 1 \times 10^{-10}$), suggesting that CHD7 shifted its predominant role from a transcriptional repressor in otic progenitors to an activator in sensory epithelia (Fig. 6c–d). GSE analysis of downregulated genes showed enrichment of hair cell differentiation and Notch signaling gene sets in hair cells, and WNT signaling and cell junction gene sets in supporting cells (Fig. 6e–h). Notably, a number of deafness genes (e.g., *SIX1*, *USH1C*, and *STRC*) from the OtoSCOPE gene panel[28] were dysregulated in *CHD7*KO/+ hair cells (Fig. 6e), providing potential explanations for a cause of hearing loss in individuals with CHARGE syndrome. Collectively, these d70 scRNA-seq results unveiled the dysregulated genes and gene sets in *CHD7*KO/+ hair cells and supporting cells bearing normal morphological properties.

In addition to hair cells and supporting cells, the inner ear organoid culture system also generates NEFL+ sensory neurons that innervate the hair cells[18,19]. In the d70 scRNA-seq dataset, *CHD7*KO/+ neurons showed minimal gene expression disruptions, with only a few ribosomal genes being downregulated (Supplementary Fig. 13a–e). Immunostaining of these mutant neurons revealed normal neurite infiltration to the sensory epithelium and normal contact with hair cells (Supplementary Fig. 13f–i), which is consistent with the normal

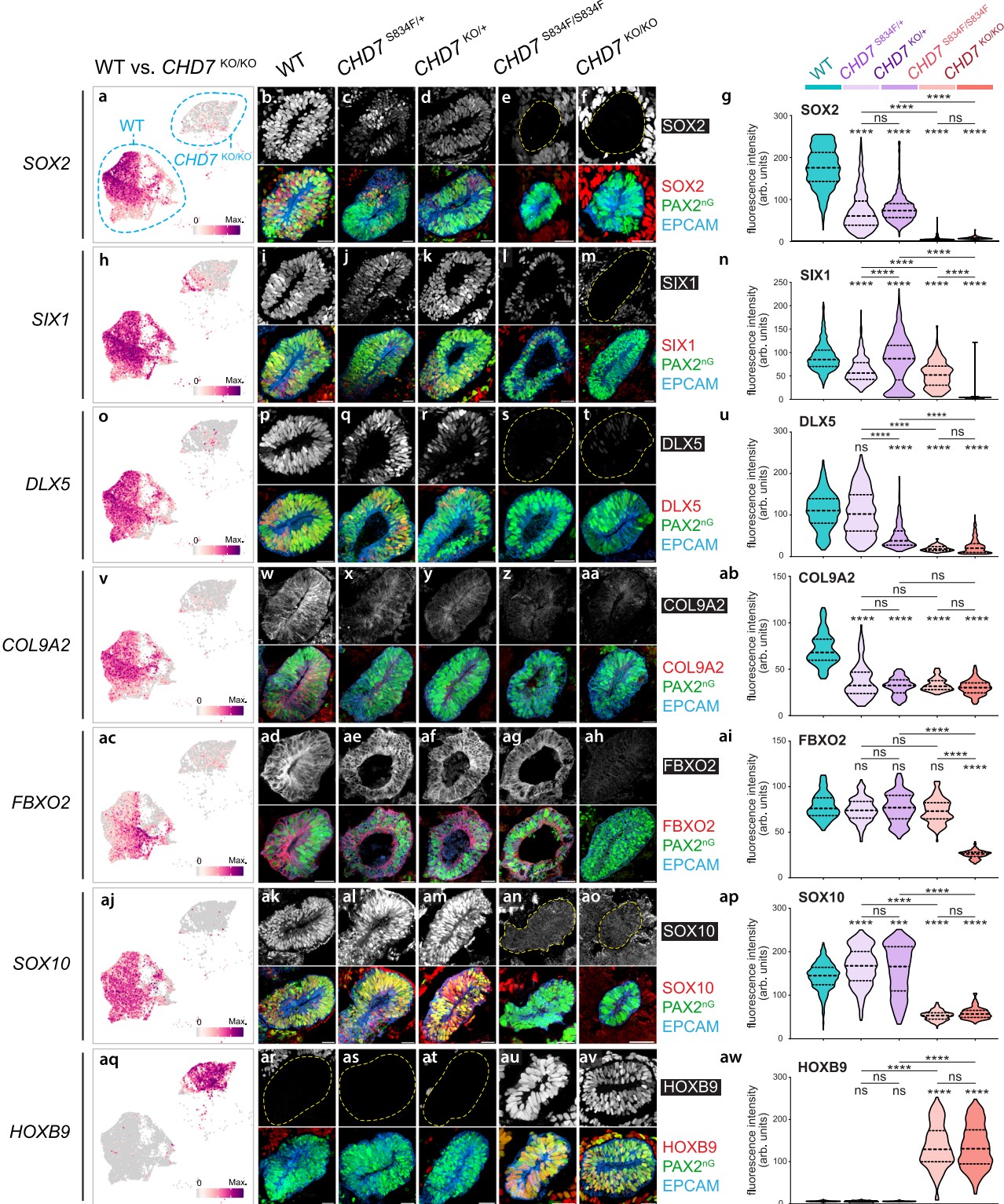

**Fig. 5 | Genes essential for otic development were dysregulated in *CHD7* mutant otic vesicles. a–aw** left panel, UMAP plots of key dysregulated genes in d20 WT and *CHD7*[KO/KO] otic progenitors. In each UMAP plot, the bottom left cluster consists mainly of WT cells (97.2%), and the top right cluster consists mainly of *CHD7*[KO/KO] cells (98.4%). The color bars to the bottom right show the log-normalized expression scale. **a–aw** middle panel, Immunostaining of SOX2, SIX1, DLX5, COL9A2, FBXO2, SOX10, or HOXB9 along with otic progenitor markers PAX2[nG] and EPCAM in WT, *CHD7*[S834F/+], *CHD7*[KO/+], *CHD7*[S834F/S834F], and *CHD7*[KO/KO] organoids. Dotted lines mark the boundaries of otic vesicles. **a–aw** right panel, Violin plot quantifications of immunofluorescence signal intensities (arbitrary units, arb. units) from each nucleus (for nuclear proteins such as SOX2 and SIX1) or

cell (for cytoplasmic proteins such as COL9A2 and FBXO2) as shown in (**a–aw** middle panel). Dashed and dotted lines indicate the median and quartile values, respectively. $n = 14{,}371$ total nuclei and cells from a total of 105 otic vesicles (136.87 nuclei or cells per otic vesicle on average). Three otic vesicles from 3 different aggregates from 3 independent experiments were used for quantification of each genotype. All PAX2[nG+] EPCAM[+] cells from each otic vesicle were quantified. ****$P < 0.0001$; ***$P < 0.001$; ns, not significant. Significance was accessed by Kruskal–Wallis test followed by Dunn's multiple comparisons test. Source data, including raw measurements and $p$ value data, are provided as a Source Data file. Scale bars, 25 μm.

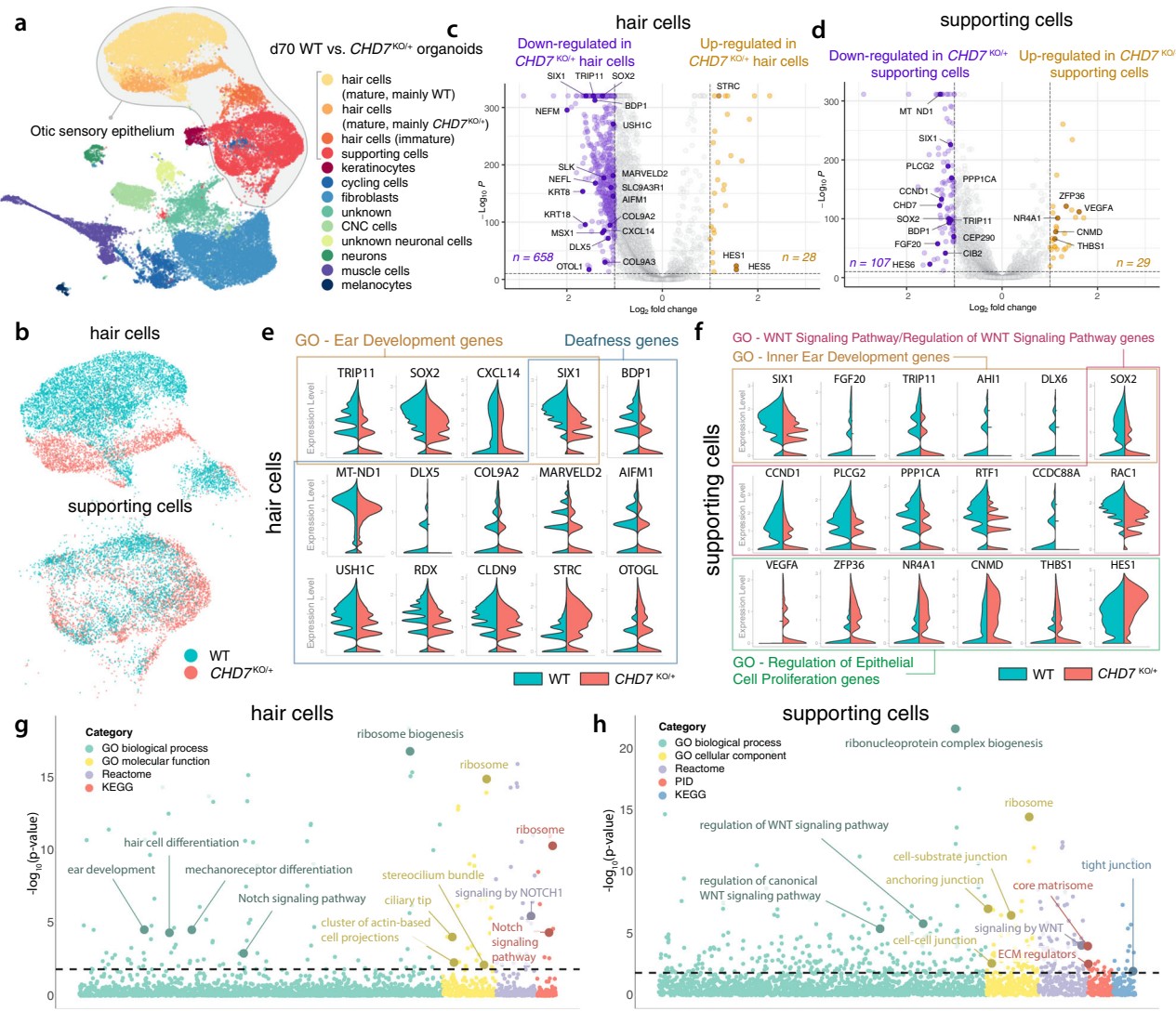

**Fig. 6 | scRNA-seq of d70 WT and *CHD7*ᴷᴼ/⁺ organoids revealed the importance of CHD7 in sensory epithelium development. a, b** d70 WT and *CHD7*ᴷᴼ/⁺ organoids were micro-dissected and FACS-isolated into *POU4F3*ⁿᵀ-positive and -negative populations. UMAP plot and cluster annotation of merged datasets of these four groups of cells were shown in (**a**), and UMAP plots of the hair cell and supporting cell subsets were shown in (**b**). Data represent 34,703 total cells (**a**), 9884 hair cells, and 6273 supporting cells (**b**). **c, d** Volcano plots of differentially expressed genes in

*CHD7*ᴷᴼ/⁺ hair cells (**c**) and supporting cells (**d**). **e, f** Split-violin plots of key genes dysregulated in *CHD7*ᴷᴼ/⁺ hair cells (**e**) and supporting cells (**f**). **g, h** Bubble plot of enriched gene sets from downregulated genes in d70 *CHD7*ᴷᴼ/⁺ hair cells (**g**) and supporting cells (**h**). For DE analysis in (**c** and **d**), the two-sided test DESeq2 was used, and adjustments were made by the Benjamini-Hochberg method for multiple comparisons. For GSEA using the one-sided iDEA platform shown in (**g** and **h**), adjustments of *p*-values were made by the Louis method.

innervation pattern found in the utricle and saccule of *Chd7* heterozygous deletion mice[8]. These data suggest that the sensory neurons appear largely unaffected by the mono-allelic loss of *CHD7*.

### *CHD7*ᴷᴼ/ᴷᴼ organoids lack hair cells and supporting cells

While the *POU4F3*ⁿᵀ⁺ and *POU4F3*ⁿᵀ⁻ FACS-based cell separation experimental design provided an adequate number of hair cells for WT and *CHD7*ᴷᴼ/⁺ organoid scRNA-seq analysis (Fig. 6 and Supplementary Fig. 11), *POU4F3*ⁿᵀ labeling for the *CHD7*ᴷᴼ/ᴷᴼ organoids was not feasible as no POU4F3⁺ hair cells were found in d70 *CHD7*ᴷᴼ/ᴷᴼ organoids (Fig. 2u–w). To enrich any potential otic epithelial populations from the *CHD7*ᴷᴼ/ᴷᴼ organoids, including *EPCAM*⁺ hair cells and *EPCAM*⁺ supporting cells, we stained dissociated d70 WT and *CHD7*ᴷᴼ/ᴷᴼ organoid cells with an EPCAM antibody, and FACS-collected *EPCAM*⁺ and *EPCAM*⁻ populations from both genotypes (Supplementary Fig. 14a–b). The merged dataset contained a single hair cell cluster composed solely of WT cells (*n* = 2810) without any *CHD7*ᴷᴼ/ᴷᴼ cells (*n* = 0) (Fig. 7),

confirming the failure of hair cell generation in *CHD7*ᴷᴼ/ᴷᴼ organoids (Fig. 2u–w).

Despite the absence of hair cells, a group of EPCAM⁺ SOX2⁺ epithelial cells encircling a luminal space were present in d70 *CHD7*ᴷᴼ/ᴷᴼ organoids. They are morphologically reminiscent of supporting cells, though their SOX2 expression levels are usually lower (Fig. 2u–w, Supplementary Figs. 15 and 16c–h). Marker gene profiling of the corresponding cluster indeed showed the expression of multiple otic lineage genes, including *FBXO2*, *OC90*, *S100A1*, *CLDN6*, and *PLEKHB1*. However, key supporting cell markers[32–35] *SPARCL1*, *BRICD5*, and *OTOG* were absent in these cells (Fig. 2u and Supplementary Figs. 16a, b, e, f, and 17), suggesting that the d70 *CHD7*ᴷᴼ/ᴷᴼ mutant organoids can give rise to otic-like epithelial tissues, but are unable to differentiate into supporting cells or hair cells. Even without hair cells or supporting cells, occasional NEFL⁺ neurite infiltration was observed in these otic-like vesicle structures, albeit at a much lower rate compared to neuron innervation in the WT and the *CHD7*ᴷᴼ/⁺ sensory epithelia (Supplementary Fig. 13j–k). Taken together, these data

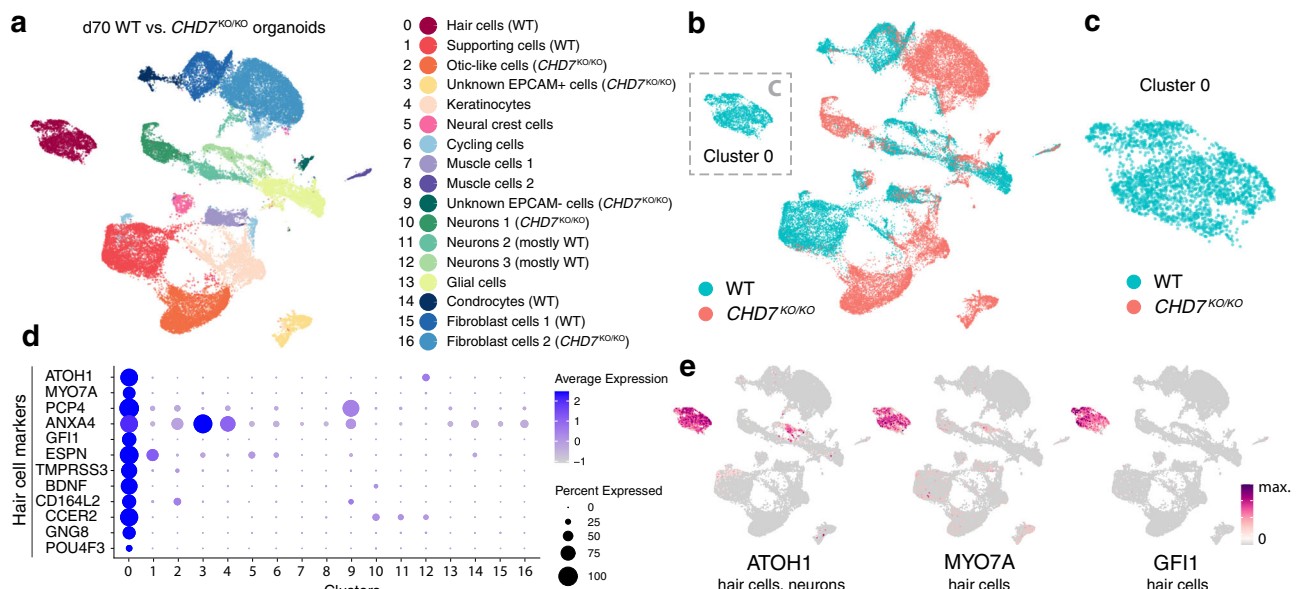

**Fig. 7 | scRNA-seq of d70 WT and *CHD7*[KO/KO] organoids suggested that CHD7 is required for hair cell generation. a–c** d70 WT and *CHD7*[KO/KO] organoids were micro-dissected and FACS-isolated into *EPCAM*-positive and -negative populations. UMAP plot and cluster annotation of merged datasets of these four groups of cells were shown in (**a**), and UMAP plot grouped by genotype were shown in (**b**).

**c** Cluster 0 subset is composed of WT cells only. Data represent 41,833 total cells (**a**, **b**), and 2810 Cluster 0 cells (**c**). **d**, **e** Dot plot of hair cell marker genes (**d**) and feature plot of hair cell markers *ATOH1*, *MYO7A*, and *GFI1* (**e**) suggested that Cluster 0 is the only hair cell cluster. The color bar in (**e**) shows the log-normalized expression scale.

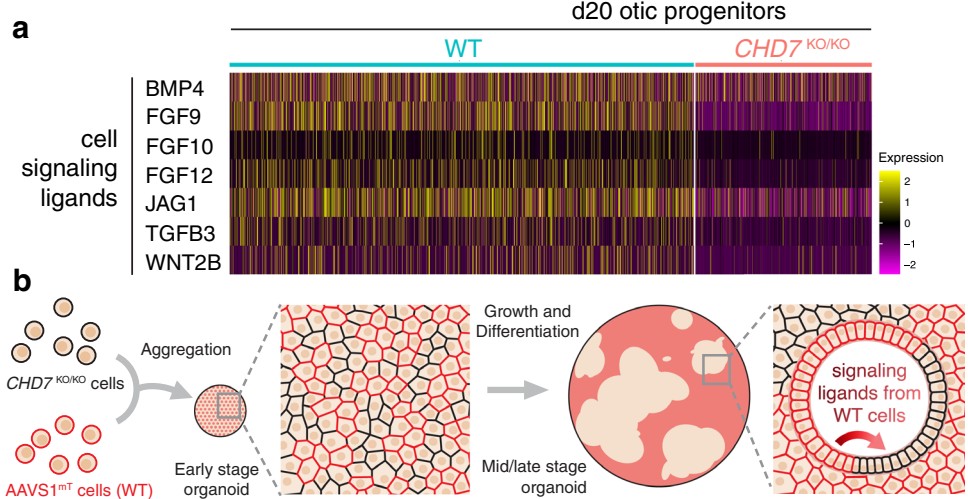

**Fig. 8 | The WT-*CHD7*[KO/KO] chimeric organoid approach is designed to test whether signaling ligands from WT cells can rescue the *CHD7*[KO/KO] phenotypes in neighboring mutant tissues. a** Heatmap showing the scaled log-normalized expression of several cells signaling ligand genes in d20 WT and *CHD7*[KO/KO] otic

progenitors. **b** WT-*CHD7*[KO/KO] chimeric organoid culture strategy for supplying *CHD7*[KO/KO] mutant cells with diffusible morphogens and cell surface signaling ligands from neighboring and nearby AAVS1[mT]-labeled WT cells.

demonstrate that CHD7 is required for hair cell and supporting cell differentiation.

**Chimeric organoids partially rescued *CHD7* mutant phenotypes**
When analyzing the d20 scRNA-seq data, we noticed that several signaling ligands, including those involved in BMP, FGF, Notch, TGFβ, and WNT signaling pathways were downregulated in the *CHD7*[KO/KO] otic progenitors (Fig. 8a). It is highly likely that dysregulation of signaling ligands also occurred in other cell types and in other developmental stages, and these abnormal levels of signaling cues likely contributed to the *CHD7* mutant phenotypes. To test this possibility, we established a chimeric organoid system by which normal levels of signaling ligands

are supplied from WT cells to the neighboring mutant cells. To distinguish between WT and mutant cells, we first labeled WT cells with a cell membrane-bound tdTomato expressed under a ubiquitous pCA promoter at the AAVS1 locus (AAVS1[mT]) (Supplementary Fig. 19). We aggregated a mixture of AAVS1[mT]-labeled WT hESCs and unlabeled *CHD7*[KO/KO] hESCs into chimeric organoids and differentiated them towards the otic lineage. As the organoids grew and were differentiated, initial single cells and small clusters expanded into larger clones, forming a mosaic WT-mutant tissue organization. In this condition, *CHD7*[KO/KO] clones received normal signaling inputs from neighboring otic and non-otic WT tissues throughout the developmental stages (Fig. 8b). Immunostaining of d20 chimeric organoids

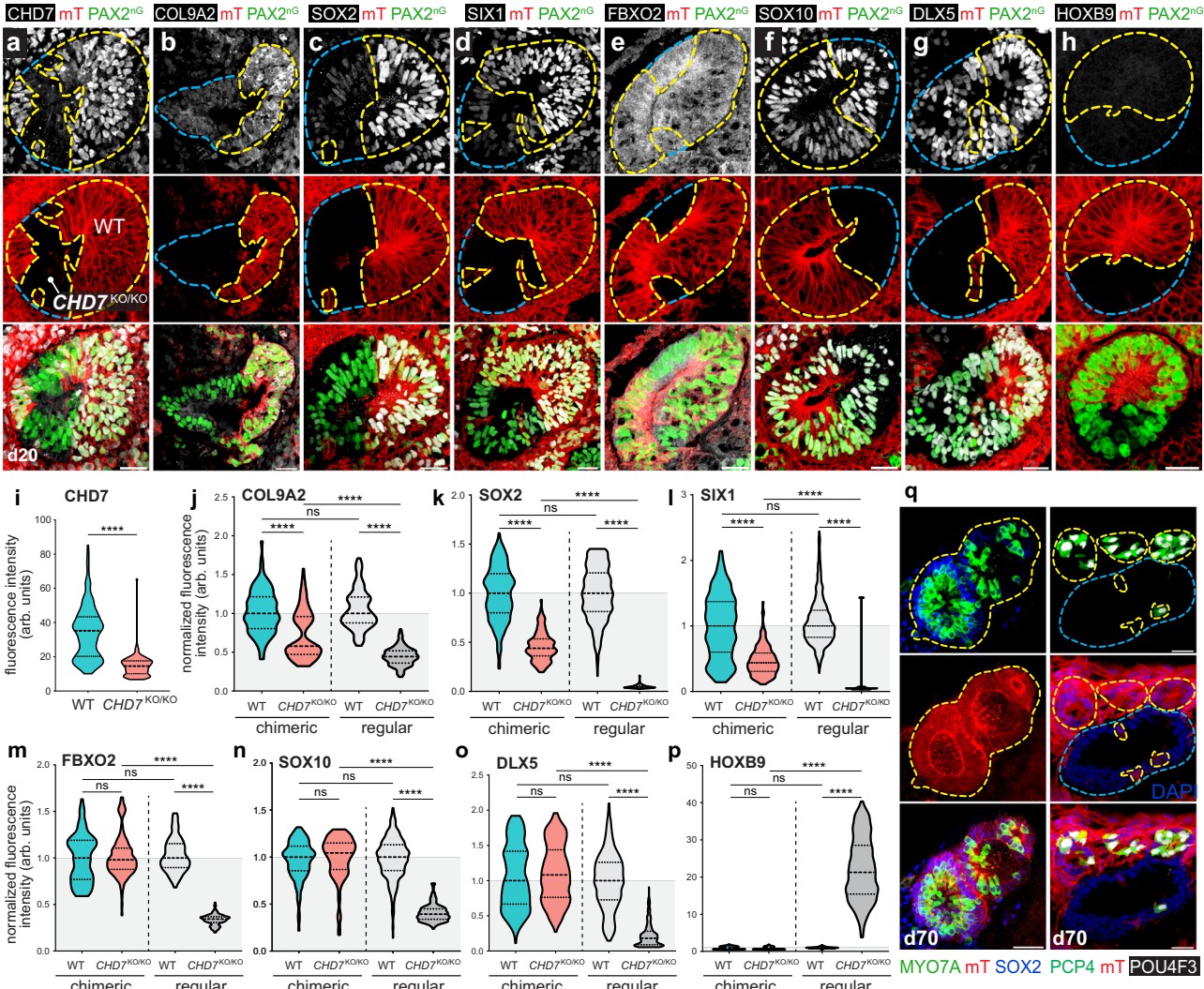

**Fig. 9 | *CHD7*[KO/KO] phenotypes in the otic progenitors can be partially rescued by co-differentiating with WT cells in chimeric organoids. a–h** Immunostaining of otic vesicles in d20 WT-*CHD7*[KO/KO] chimeric organoids. Yellow dotted lines highlight the AAVS1[mT]-labeled WT clones in an otic vesicle, and the blue dotted lines highlight the *CHD7*[KO/KO] otic vesicle clones. **i–p** Violin plot quantifications of immuno-fluorescence intensity data (arbitrary units, arb. units) as shown in (**a–h**). WT and *CHD7*[KO/KO] data from chimeric organoids were normalized to the chimeric organoid WT cell median values, and were shown in blue and red, respectively. Regular single-genotype WT and *CHD7*[KO/KO] organoids data from (Fig. 5) were normalized to regular WT organoid median values, and were shown in light and dark gray, respectively. Dashed and dotted lines indicate the median and quartile values, respectively. *n* = 3153 total nuclei and cells from a total of 24 otic vesicles (131.38 nuclei or cells per otic vesicle on average). A total of three otic vesicles from three different aggregates from three independent experiments were used for quantifi-cation of each genotype. All PAX2[nG+] EPCAM[+] WT and *CHD7*[KO/KO] cells from each otic vesicle were quantified. **q** hair cells in d70 WT-*CHD7*[KO/KO] chimeric organoids labeled with MYO7A, PCP4, POU4F3, and SOX2. Yellow dotted lines highlight the AAVS1[mT]-labeled WT clones in the sensory epithelia, and the blue dotted lines highlight the *CHD7*[KO/KO] clones. ****P < 0.0001; ns, not significant. Significance was accessed by Kruskal–Wallis test followed by Dunn's multiple comparisons test. Source data, including raw measurements and *p* value data, are provided as a Source Data file. Scale bars, 25 μm.

with a CHD7 antibody confirmed the expected mosaic pattern of mT[+] CHD7[+] WT tissues and mT[−] CHD7[−] mutant tissues (Fig. 9a, i). Remarkably, under this chimeric culture condition, the otherwise severely dysregulated FBXO2, SOX10, DLX5, and HOXB9 proteins were restored comparable to the WT level (Fig. 9e–h, m–p). As the *CHD7* mutant phenotypes most likely resulted from both aberrant extrinsic signaling inputs and dysregulated intrinsic gene expression networks, we did not anticipate this strategy to provide a full rescue for all affected genes. Consistent with this expectation, we observed partial rescues of COL9A2, SOX2, and SIX1 (Fig. 9b–d, j–l). At d70, all derived hair cells were mT[+] WT cells, and no mT[−] *CHD7* null mutant hair cells were observed in chimeric organoids (*n* = 180 aggregates in 12 inde-pendent chimeric organoid cultures) (Fig. 9q). The failure of *CHD7*[KO/KO] hair cell generation likely resulted from the incomplete rescue of key otic genes such as COL9A2, SOX2, and SIX1 at earlier developmental stages (Fig. 9b–d, j–l). Collectively, these results demonstrate that the *CHD7* KO phenotypes can be partially rescued, and that the drifted otic lineage identity can be partially restored in the otic progenitors in chimeric organoid cultures.

## Discussion

In this study, we recapitulated the pathogenesis of CHARGE syndrome with human inner ear organoids as a model system. Previous mouse studies did not allow for investigation of mature inner ear phenotypes associated with homozygous *Chd7* mutation, as these mice survive only up to embryonic day 10.5 (E10.5), by which time the otic lineage cells only developed to the otic progenitor stage. Conditional KO of *Chd7* in otic lineage cells has been used as an alternative approach, but none of the Cre recombinases used in these experiments were expressed before E8.5[9–11,13,15,16], making it difficult to assess the effect of

*Chd7* deficiency initiated earlier developmental stages. While the heterozygous *CHD7* mutant phenotypes uncovered in previous studies are highly clinically relevant due to the prevalence of heterozygosity in CHARGE clinical cases, investigation into the homozygotes is scientifically critical for gaining insights into a comprehensive downstream gene network and the full-scale developmental significance of CHD7. In the stem cell-derived inner ear organoid system, the survival and development of otic lineage cells are not dependent on the proper development of vital organs such as the heart, making organoids a suitable platform to study embryonically lethal genes. Using this approach, we revealed that genetic ablation of *CHD7* or its chromatin remodeling activity leads to dysregulation of early otic lineage genes and a partially drifted otic lineage identity, resulting in a complete absence of hair cells and supporting cells.

By epigenetically altering the chromatin architecture to modulate the nucleosome accessibility, along with other less understood mechanism(s), CHD7 exerts transcriptional control of hundreds of tissue-specific downstream genes[36-39]. One of the *CHD7* otic target genes, *SOX2*, appears to be one of the leading causes. We identified *SOX2* as one of the top differentially expressed genes following *CHD7* deletion. *SOX2* expression in otic progenitors is completely abolished in the bi-allelic *CHD7* KO or S834F mutants, and the corresponding mono-allelic *CHD7* mutations lead to reduced *SOX2* expression. Sox2 is known to activate the hair cell initiator gene *Atoh1*[40-42], and loss of *Sox2* expression in mouse otic tissues results in failure of hair cell and supporting cell generation[43], which phenocopies the absence of hair cell and supporting cells in *CHD7* null organoids. Moreover, reduced *SOX2* expression leads to truncated semicircular canals, shortened cochleae, and hearing impairment[43,44], which are also often observed in heterozygous *Chd7* mutant mice and individuals with CHARGE syndrome[8,45-47]. In addition to being a *CHD7* downstream gene, Sox2 has been reported to physically interact with Chd7 and co-occupy genomic binding sites to regulate common target genes[48]. Therefore, the reduction of SOX2 expression in *CHD7* mutants may further dysregulate genes downstream of the CHD7-SOX2 complex. Although the downregulation of *SOX2* alone accounts for many of the *CHD7* mutant phenotypes, it should be noted that *SOX2* and *CHD7* play distinct roles during inner ear development, and their mutant phenotype spectrums do not completely overlap[11,41,43,44,49-52]. Therefore, it is almost certain that the combined effects of many dysregulated *CHD7* downstream genes are responsible for the CHARGE inner ear phenotypes.

Previous studies have shown that heterozygous *Chd7* deficient mice had stereocilia-bearing hair cells that are morphologically indistinguishable from the WT counterpart[8], but it is unclear whether the gene expression profile or the function of hair cells is affected in these mutant mice. Using single-cell transcriptome profiling, we demonstrate that multiple deafness genes from the OtoSCOPE panel[28], including *SIX1*, *USH1C*, and *CLDN9*, were dysregulated in the mono-allelic *CHD7* KO organoid hair cells. These results suggest that the dysregulation of deafness genes in *CHD7*[+/-] hair cells may be one of the underlying mechanisms of CHARGE syndrome-associated hearing loss and balance dysfunction, which are accompanied by middle and inner ear malformations and neurogenic defects as revealed by previous mouse model and human patient studies[8,9,46,50]. It should be noted that hair cells generated with our current organoid differentiation protocol exhibit electrophysiological, morphological, and marker gene expression characteristics of vestibular hair cells (Supplementary Fig. 12)[18-20]. We hypothesize that the deafness genes identified in this vestibular in vitro model are essential for the mechanotransduction function in both hearing and vestibular organs. Consistent with this hypothesis, *CHD7* and all *CHD7*-regulated deafness genes identified in this study are expressed in both cochlea and vestibule during mouse development (Supplementary Fig. 20)[9,34]. Moreover, 67% of the deafness genes identified in this study are also known to be associated with vestibular dysfunctions and defects (Supplementary Table 2), despite

the fact that genetic causes of vestibular disorders are less well studied than hearing loss, leading to likely underreported vestibular dysfunction genes and the lack of comprehensive vestibular dysfunction gene panels[53]. Therefore, we believe that the deafness genes identified in this vestibular organoid system could serve as good candidate genes for future studies in cochlear *CHD7* disease models.

CHD7 is known as an ATP-dependent chromatin remodeling enzyme[1]. The S834F mutation has been shown to completely abolish CHD7's ATPase and chromatin remodeling activities[2], raising the possibility that this patient-specific missense mutant protein represents a functional null. Indeed, we observed similar cell morphological phenotypes between the S834F mutant and its corresponding mono- or bi-allelic KO mutants, and in most cases, similar downstream gene expression levels between these two types of mutants. However, there are notable exceptions. For example, the highly specific otic lineage marker FBXO2 was severely downregulated in the bi-allelic KO, but its expression was maintained at the WT level in the bi-allelic S834F mutant. These results imply the presence of additional CHD7 protein function(s) beyond its ATP-dependent chromatin remodeling activities, and such function(s) play critical roles in regulating the expression of some of the key otic genes. Recently, a chromatin remodeling-independent function of CHD7 was identified in mouse cardiovascular development, where CHD7 binds to WDR5, a core component of an H3K4 methyltransferase complex, to recruit this histone-modifying enzyme complex to its target gene loci to regulate gene expression[54]. Future studies are needed to test if similar chromatin remodeling-independent functions of CHD7 are responsible for regulating otic developmental genes such as *FBXO2*, *SIX1*, and *DLX5* in the inner ear.

Genetic chimeric animals can be generated through genomic integration of complicated sets of gene circuits as seen in the *Drosophila* MARCM mosaic tissue generation system[55] and in the mouse MADM system[56]. Chimeric animals can also be generated by simply mixing WT and mutant mouse embryonic stem cells in the blastocysts at the preimplantation stage[57]. Inspired by these models, we have established a chimeric organoid culture system to analyze cell-autonomous and non-autonomous actions by mixing mutant and fluorescently labeled WT embryonic stem cells. Through co-differentiating, these cells into chimeric organoids, the expression levels of otherwise dysregulated genes in the *CHD7* KO otic progenitors were fully or partially restored. This can be explained by diffusible morphogens or cell surface signaling ligands provided from developing WT cells to the neighboring co-developing mutant cells during the co-differentiation process. The complete rescue of *FBXO2*, *SOX10*, *DLX5*, and *HOXB9* and the partial rescue of *COL9A2*, *SOX2*, and *SIX1* genes indicate that paracrine signaling from WT cells helped the *CHD7* KO cells to develop more faithfully the otic lineage path, thereby restoring the expression of these otic genes to full or partial extents. It will be interesting to test if the paracrine signaling ligands supplementation can rescue *CHD7* phenotypes in other organs, or alleviate phenotypes of genetic diseases caused by other genes.

The human inner ear organoids serve as a valuable model for accelerating our understanding of human-specific aspects of the embryonic lethal gene *CHD7*, yet it is also important to recognize the limitations of this in vitro approach. First, as mentioned above, the organoid differentiation protocol used in this study generated only vestibular hair cells but not cochlear hair cells (Supplementary Fig. 12)[18-20], thus the functions of *CHD7* in cochlear hair cells and supporting cells cannot be directly inferred from this study. In addition, the effects of *CHD7* on morphogenesis of the middle ear or the inner ear structures, such as semicircular canals and the cochlear duct, cannot be recapitulated in the current system. Also, influences from vasculature, immune cells, and morphogen gradients established by neighboring tissues such as the neural tube and notochord are lacking. Additionally, like most of the current organoid models, cells in human inner ear organoids do not fully mature and represent immature fetal

developmental stages even after prolonged culture. Nevertheless, a full understanding of human disease can be achieved only from comprehensive knowledge gained from various in vitro and animal models, as well as human patient clinical studies.

In summary, we demonstrate the critical role of *CHD7* in regulating human otic lineage differentiation and deafness gene expression. Loss of CHD7 or its ATP-dependent chromatin remodeling function results in a failure of hair cells and supporting cell generation in human inner ear organoids. The differential expression of a subset of *CHD7* downstream genes between S834F and its corresponding mono- or bi-allelic KO mutant suggests the presence of CHD7 functions beyond its ATP-dependent chromatin remodeling activities. Notably, the aberrant otic lineage identity and the dysregulation of key otic genes in *CHD7* KO cells can be partially rescued by co-culturing and co-differentiating with WT cells in a mosaic manner, highlighting the contribution of abnormal extrinsic signaling inputs in the *CHD7* mutant phenotypes. Our findings shed light on the molecular basis of inner ear phenotypes associated with CHARGE syndrome and reveal potential therapeutic target genes and pathways. The various human *CHD7* mutant lines established in this study will be valuable resources for future drug screening and validating human genome-specific gene therapy approaches.

## Methods

### hESC culture

Human ESCs (WA25 hESCs and genomically edited cell lines based on the WA25 background, passage 13–55) were cultured in Essential 8 Flex (E8f) medium (Thermo Fisher) supplemented with 100 μg/mL Normocin (Invivogen) (hereafter, E8fn medium) on truncated recombinant human Vitronectin-N (Thermo Fisher)-coated Nunclon Delta surface-treated 6-well plates (Thermo Fisher) according to an established protocol[58]. At 60%–80% confluency or every 3 or 4 days, the cells were passaged at a split ratio of 1:100–1:10 into 2–4 wells of a 6-well plate using 0.5 mM Ethylenediaminetetraacetic acid (EDTA) in Dulbecco's phosphate-buffered saline (DPBS). 1X RevitaCell (Thermo Fisher) was supplemented in the E8fn medium for 1 d after passaging for increased viability. The WA25 hES cell line was acquired from the WiCell Research Institute. For additional validation and testing information, refer to the cell line webpage: https://www.wicell.org/home/stem-cell-lines/catalog-of-stem-cell-lines/wa25.cmsx.

### CRISPR genomic editing

To generate the *CHD7*–3 × Flag cell line, a *CHD7*–3 × Flag-PGK-Puro donor plasmid was constructed by cloning the following fragments into pUC19 plasmid backbone using Gibson Assembly[59]: left and right homology arms (LHA and RHA) homologous to 1 kb genomic DNA upstream and downstream of the *CHD7* stop codon locus, respectively (PCR amplified from WA25 genomic DNA), Glycine-Serine linker-3 × Flag-STOP-bGH polyA (gBlock DNA, IDT), and a PGK-Puro cassette (Addgene #31938)[60]. Ribonuclease protein (RNP) complex[61] targeting the *CHD7* stop codon locus was assembled by incubating high fidelity Cas9 protein (HiFi Cas9 nuclease 3NLS, IDT) with crRNA:tracrRNA duplex (*CHD7* crRNA: 5′– ACTTGAACTGGAACTGGTAC –3′, IDT). The RNP complex, the donor plasmid, as well as an electroporation enhancer (IDT) were transfected into the WA25-based *PAX2*–2A-nGFP reporter hES cell line[26] with 4D Nucleofector (Lonza) using the P3 Primary Cell 4D-Nucleofector X kit (Lonza) and Program CB-150. After nucleofection, cells were plated in RevitaCell-supplemented E8fn medium for one day for improved cell survival rate, followed by treatment with 1 μM Scr7 (Xcessbio) on the second day for enhanced HDR efficiency[62]. A total of 0.25–0.5 μg/mL puromycin (Thermo Fisher) selection was performed for four days starting from the third-day post-nucleofection. Clonal cell lines were established by low-density seeding followed by isolation of hESC colonies after 6 days of expansion. Genotypes of cell lines were analyzed by PCR and Sanger sequencing.

Cell lines with bi-allelically inserted 3×Flag at the *CHD7* stop codon locus were used for downstream validation and experiments.

*CHD7* mono-allelic and bi-allelic knockout (KO) cell lines were generated with the double-nicking CRISPR strategy[63] to minimize any potential off-target mutation. Two gRNAs (5′–GGGGTGTGATACTG CGAGTG –3′ and 5′– GTTCCTCAGGTGCCCCATGG –3′, offset = 4 bp) targeting the first coding exon of *CHD7* were individually cloned into a gRNA expression plasmid pSPgRNA (Addgene #47108)[64]. The two gRNA plasmids, along with a double-nicking Cas9 expression plasmid hCas9_D10A (Addgene #41816)[65] and a puromycin expression plasmid pPGKpuro (Addgene #11349)[66] were transfected into the WA25-based *PAX2*–2A-nGFP reporter hES cell line with a 4D Nucleofector. After nucleofection, cells were plated in RevitaCell-supplemented E8fn medium for one day for improved cell survival. To enrich transfected cells, 0.5 μg/mL puromycin selection was performed for two days starting from 48 h post-nucleofection. After confirming successful indel formation at the cell population level using the T7 endonuclease 1 assay (T7E1, New England Biolabs), clonal cell lines were established by low-density seeding (1–3 cells/cm²) of accutase (Thermo Fisher)-dissociated single cells followed by isolation of hESC colonies after 5–7 d of expansion. T7E1 assay showed 26 out of 94 (28%) established clonal cell lines contained indels. Next-generation sequencing (NGS) of amplicon DNA was performed on the 26 T7E1-positive cell lines at the *CHD7* first coding exon locus (Genome Engineering and iPSC Center, Washington University). Among these cell lines, 5 cell lines (19%) had frame-shift indels on both alleles (bi-allelic mutant cell lines, indel size ranging from −25 bp to +79 bp), 13 cell lines (50%) had frame-shift indels on one allele and wild-type (WT) sequence on the other allele (mono-allelic mutant cell lines, indel size ranging from −20 bp to +38 bp), and the remaining 8 cell lines contained in-frame indels on one or both alleles (31%). A bi-allelic cell line with a 25 bp deletion and an 8 bp insertion, as well as a mono-allelic cell line with an 11 bp deletion and a WT allele were selected for further validation and analysis. To test whether each of these two cell lines were homogenous cell lines or mixtures of cells with different genotypes, multiple second-round clonal cell lines were derived from each of these two parental lines. TIDE (Tracking of Indels by Decomposition) analysis[67] of the second-round clonal cell lines showed identical indel patterns as their parental lines, with ~50% frequencies for both of the two peaks, thus confirming that these two *CHD7* mutant cell lines were homogenous.

To generate mono-allelic and bi-allelic *CHD7* S834F missense point mutation hESC lines, a gRNA (5′–GCTCTTATCTTCATTG TCAG–3′) targeting the *CHD7* p.S834 c.2501 C locus was cloned into the gRNA expression plasmid pSPgRNA (Addgene #47108)[64]. An optimized CRISPR base editor expression cassette (BE-FNLS-2a-Puro) was sub-cloned from its original lentiviral vector (Addgene #110841)[25] into a pUC19 vector backbone. These two plasmids were nucleofected into the WA25-based *PAX2*–2A-nGFP reporter hES cell line followed by 0.5 μg/mL puromycin selection on the following day. After a cell passaging on post-nucleofection day 7, genomic DNA from an aliquot of day 13 targeted population of cells was harvested for PCR and sequencing. The genotyping results demonstrated ~44% C to T conversion rate as analyzed by the EditR tool[68]. A total of 48 clonal cell lines were established and screened with PCR and Sanger sequencing, in which 27 cell lines contained a total of 44 c.2501 C > T p.S834F mutant alleles. After excluding cell lines with C > T conversions at c.2506 C, cell lines with indels, and cell lines showing signs of heterogeneity, we obtained seven bi-allelic and five mono-allelic c.2501 C > T p.S834F mutant cell lines. One cell line from each of the two genotypes was used for downstream validation and experiments.

To generate the *POU4F3*–2a-ntdTomato (*POU4F3*nT) *CHD7*KO/+ cell line, we constructed a pUC19-*POU4F3*–2a-tdTomato-nls-bGHpA donor plasmid via restriction enzyme digestion and T4 ligase-based subcloning and Gibson assembly. RNP complex targeting the *POU4F3* stop codon locus was assembled by incubating high fidelity Cas9 protein

(HiFi Cas9 nuclease 3NLS v3, IDT) with a *POU4F3* sgRNA (5′–ATTCGGCTGTCCACTGATTG–3′, Synthego). The RNP complex, the donor plasmid, a puromycin expression plasmid pPGKpuro (Addgene #11349)[66], and an electroporation enhancer (IDT) were transfected into the established *CHD7*KO/+ parental hESC line by nucleofection. After nucleofection, cells were plated in RevitaCell-supplemented E8fn medium for one day, followed by treatment with 1 μM Scr7 on the second day for enhanced HDR efficiency[62]. A total of 0.25 μg/mL puromycin selection for transfected cells was performed on the second and the third day after nucleofection. Clonal cell lines were established by low-density seeding followed by isolation of hESC colonies. Genotypes of cell lines were analyzed by PCR and Sanger sequencing. Cell lines with bi-allelically inserted 2a-ntdTomato at the *POU4F3* stop codon locus were used for downstream validation and experiments.

To generate the AAVS1-mT (membrane-localized tdTomato) cell line, a AAVS1-pCA-mT donor plasmid was constructed by cloning 1 kb AAVS1 LHA and RHA (PCR amplified from WA25 genomic DNA) and pCA-mT (Addgene #17787)[69] into pUC19 plasmid backbone using Gibson Assembly. The donor plasmid was nucleofected into WA25 hESCs along with a high fidelity Cas9 RNP complex (AAVS1 crRNA: 5′–ACCCCACAGTGGGGCCACTA –3′), an electroporation enhancer, and a pPGKpuro plasmid. RevitaCell, 1 μM Scr7, and 0.5 μg/mL puromycin were used to treat the transfected population of cells, and low-density cell seeding was performed as described above. As the ubiquitous pCA promoter drives the expression of mT in all cell types at all developmental stages, including in hESCs, only hESC colonies emitting the mT fluorescence signals were isolated to establish clonal cell lines. PCR and Sanger sequencing were used to confirm successful mT knockin at the AAVS1 locus in these cell lines.

For all CRISPR genomically engineered hES cell lines, pluripotency was verified by immunohistochemistry of pluripotency markers (OCT4, SSEA4, and SOX2), and normal hESC colony morphologies were verified with a bright-field microscope. Top 10 predicted off-target sites (www.crispr.mit.edu and www.guidescan.com[70]) of each gRNA were PCR amplified (~1 kb) from the genomic DNA of established cell lines and were Sanger sequenced to test for off-target mutations. Karyotyping assays were performed at KaryoLogic, Inc.

## Human inner ear organoid culture

Human inner ear organoids were derived from hESCs following our previous protocol[19] with modifications. Briefly, to start differentiation, hESCs cultured on 6-well plates were washed three times with DPBS (Thermo Fisher) followed by dissociation with accutase (Thermo Fisher) for 8 min at 37 °C. Dissociated cells were pelleted by centrifuging for 3 min at $100 \times g$ and were resuspended in E8fn medium containing 20 μM Y-27632 (Stemcell Technologies) to a final concentration of 35,000 cells/mL. Hundred microlitres of cells were added to each well (3500 cells per well) of Nunclon Sphera low-binding 96-well U-bottom plate(s) (Thermo Fisher) and were centrifuged to the bottom of the wells at $120 \times g$ for 5 min. After ≥4 h of incubation at 37 °C 5% $CO_2$, 100 μL of E8fn were added to each well to decrease the concentration of Y-27632 to 10 μM. Following a 48 h incubation after cell seeding, the aggregates were transferred to fresh low-binding 96U plates in 180 μL of chemically defined medium (CDM) containing 2% Matrigel (Corning), 10 μM SB-431542 (Stemcell Technologies), 4 ng/mL FGF-2 (Stemcell Technologies), and 100 pg/mL BMP-4 (ReproCell) to initiate non-neural induction – that is, differentiation day 0 (d0). CDM contained a 1:1 mixture of F-12 Nutrient Mixture with GlutaMAX (Thermo Fisher) and Iscove's Modified Dulbecco's Medium with GlutaMAX (IMDM; Thermo Fisher), additionally supplemented with 0.5% Bovine Serum Albumin (BSA, Sigma), 1× Chemically Defined Lipid Concentrate (Thermo Fisher), 7 μg/mL Insulin (Sigma), 15 μg/mL Transferrin (Sigma), 450 μM Mono-Thioglycerol (Sigma), and 100 μg/mL Normocin (Invivogen). On day 4 of differentiation culture, 45 μL of CDM containing 250 ng/mL FGF-2 (50 ng/mL final concentration) and

1 μM LDN-193189 (200 nM final concentration; ReproCell) was added to the pre-existing 180 μL of media in each well. On day 8 of differentiation culture, 45 μL of CDM containing 18 μM CHIR-99021 (3 μM final concentration; Stemcell Technologies) was added to the pre-existing 225 μl of media in each well. On differentiation day 11, the aggregates were pooled together, washed with DMEM:F12 with HEPES (Thermo Fisher), and resuspended in freshly prepared Organoid Maturation Medium (OMM) supplemented with 1% Matrigel and 3 μM CHIR-99021. The OMM medium contains a 1:1 mixture of Advanced DMEM:F12 (Thermo Fisher) and Neurobasal Medium (Thermo Fisher) supplemented with 0.5× N2 Supplement (Thermo Fisher), 0.5× B27 without Vitamin A (Thermo Fisher), 1× GlutaMAX (Thermo Fisher), 0.1 mM β-Mercaptoethanol (Thermo Fisher), and 100 μg/mL Normocin. Starting from day 11, the aggregates were cultured stationary on non-coated 100 mm dishes. On day 13 and day 15, medium was changed with OMM supplemented with 3 μM CHIR-99021. Starting on day 18, the aggregates were cultured in OMM without additional supplements. OMM medium change were performed twice a week or when the color of the medium start to turn orange or slightly yellow. Aggregates can be gently washed off the dishes and transferred to new 100 mm dishes during medium change to get rid of the migrating cells growing adherently on the dishes, which may compete with the aggregates to consume the culture medium.

To start differentiation of the WT-*CHD7*KO/KO chimeric cultures, WT (AAVS1mT) and *CHD7*KO/KO hESCs were separately dissociated and resuspended to a final concentration of 35,000 cells/mL. An equal volume of WT and *CHD7*KO/KO hESCs were mixed in a fresh tube, followed by seeding of 100 μL of the cell mixture to each well of low-binding 96U plates (3500 cells per well). The rest of the culture procedures were the same as regular organoid cultures.

## Organoid dissociation and scRNA-seq

For d20 WT vs. *CHD7*KO/KO scRNA-seq, 30 WT aggregates (*PAX2*nG) and 30 *CHD7*KO/KO aggregates (*CHD7*KO/KO *PAX2*nG) (as well as 10 WA25 control aggregates dissociated in separate tubes and wells) were washed three times with DPBS, three times with 1.1 mM EDTA, followed by resuspension in an accutase solution. A total of 10–15 aggregates were transferred to each well of a Nunclon Sphera low-binding 6-well plate (Thermo Fisher) along with 3 mL of accutase. The plates were incubated at 37 °C 5% $CO_2$ for 90 min with gentle trituration every 5–10 min with a wide-bore P1000 pipet tip. Dissociated cells were filtered through a 100 μm cell strainer and then a 40 μm cell strainer (Corning), and then centrifuged in 2 mL round-bottom tubes at $100 \times g$ for 3 min. Cell pellets were resuspended in a DMEM:F12 solution (with HEPES, no phenol red; Thermo Fisher) supplemented with 10% FBS (Thermo Fisher) and 1:500 propidium iodide (Thermo Fisher) cell viability dye. GFP+ propidium iodide- cells were sorted into a DMEM:F12 (with HEPES, no phenol red) solution supplemented with 10% FBS on a SORP Aria FACS machine (BD Biosciences) for 1 h at Indiana University Flow Cytometry Resource Facility, using dissociated cells from d20 WA25 aggregates as a negative control for gating.

For d20 WT *PAX2*nG+ and *PAX2*nG− scRNA-seq, the cell dissociation, and FACS sorting were performed similarly, with the exception that 1:500 7-AAD (BioLegend) were used as a viability dye, and that both GFP+ and GFP− populations were separately collected for downstream scRNA-seq experiments.

To enrich hair cells and supporting cells from d70 WT (*POU4F3*nT *PAX2*nG) and *CHD7*KO/+ (*CHD7*KO/+ *POU4F3*nT *PAX2*nG) organoids, tissues containing vesicle structures harboring the *POU4F3*nT-positive hair cells were micro-dissected from the rest of the d70 aggregates with fine tweezers (Dumont) under a fluorescence stereomicroscope. The dissected d70 organoid tissues were dissociated and FACS sorted in a similar way as d20 organoids, with the exception that no viability dye was used. tdTomato+ and tdTomato− cell populations were collected in separate tubes for separate downstream scRNA-seq reactions.

For d70 WT (*POU4F3*[nT] *PAX2*[nG]) vs. *CHD7*[KO/KO] (*CHD7*[KO/KO] *PAX2*[nG]) scRNA-seq, the *POU4F3*[nT] FACS sorting strategy can no longer be used to enrich the otic epithelial cell types, as the *CHD7*[KO/KO] organoids do not generate any *POU4F3*[nT+] hair cells. Therefore, while *POU4F3*[nT] fluorescence signals were still used as a guide for micro-dissection for the WT organoids, phase contrast live imaging was used for micro-dissection of the *CHD7*[KO/KO] organoids to enrich tissues containing vesicle morphological structures. After tissue dissociation, cells were stained with 1:100 PE-conjugated EPCAM antibody (BioLegend) by nutating at 4 °C for 40 min in the dark. No viability dye was co-stained. After antibody staining, cells were washed twice prior to FACS sorting. Both PE-EPCAM+ and PE-EPCAM− populations were collected in separate tubes for separate downstream scRNA-seq reactions. For WT samples, as the *POU4F3*[nT+] hair cells also express EPCAM on their cell surface, the hair cells exhibit both tdTomato and PE fluorescence signals, both of which are red. As such, the WT hair cells were FACS-isolated as a sub-population of PE-EPCAM+ cell. The *POU4F3*[nT] reporter is highly specific to the EPCAM-expressing hair cells, and the PE-EPCAM− population does not contain any *POU4F3*[nT] signals.

Sorted single cells were captured, lysed, and cDNA libraries were generated using a Chromium Controller and Single Cell 3′ Reagent Kits V3 (10X Genomics) following the manufacturer's instructions. cDNA library quality was verified using a bioanalyzer (Agilent Technologies), followed by sequencing using the NovaSeq 6000 sequencing system (Illumina) at Indiana University School of Medicine Center for Medical Genomics.

### scRNA-seq data analysis
Illumina's Real Time Analysis software was used to generate a BCL file, which was subsequently de-multiplexed and converted to a FASTQ file by the bcl2fastq Conversion Software (Illumina). The Cell Ranger pipeline was used to process the FASTQ file as follows: De-multiplexed reads were mapped to the GRCh38/hg38 human reference genome with the STAR (Spliced Transcripts Alignment to a Reference) aligner, mapped reads were grouped by cell barcode, and single-cell gene expression was quantified using unique molecular identifiers (UMIs). The resulting filtered gene-barcode (count) matrix was used as input for downstream analysis.

Using the Seurat[71] v4.0.3 R package, scRNA-seq datasets were loaded to R and converted to Seurat objects using Seurat functions Read10X and CreateSeuratObject, respectively. Low-quality cells with an extremely high or low numbers of detected UMIs and cells with high percentage of mitochondrial reads were filtered out from subsequent analysis. Datasets were merged across samples, followed by data normalization, scaling, and variable gene identification using the SCTransform function. Principal component analysis (PCA) was performed and the first 30 principal components were retained for downstream analysis. Clustering was performed with FindNeighbors and FindClusters functions, and cluster markers were identified using the FindMarkers function in Seurat.

Differential expression (DE) analysis was performed using the DESeq2 package[72] in conjunction with zingeR in R, and DE genes were visualized on volcano plots using the EnhancedVolcano R package (https://github.com/kevinblighe/EnhancedVolcano). To generate the bubble plot for differentially expressed otic-specific genes, the gene expression fold-change values of E10.5 otic vesicles versus non-otic tissues reported by Hartman et al.[30] were used to calculate the area of gene-correlated spots on the volcano plot.

Gene set enrichment (GSE) analysis was performed using the Integrative Differential expression and gene set Enrichment Analysis (iDEA) R package[29]. The DESeq2 differential expression analysis results were used as a summary statistics input for iDEA, and upregulated genes and downregulated genes were analyzed separately with iDEA. All gene sets used for GSEA, some of which were included with the iDEA

package[29], can be downloaded from the MSigDB database (http://www.gsea-msigdb.org/gsea/msigdb/collections.jsp).

### Immunohistochemistry
Aggregates were fixed with 4% paraformaldehyde (PFA, Electron Microscopy Sciences) for 30 min at room temperature (RT) followed by graded treatment of 15% and 30% sucrose and embedding in tissue-freezing medium. Frozen tissue blocks were sectioned into 10–20 μm cryosections on a Leica CM-1860 cryostat. For the fixation of hESCs, cells growing on the 6-well plates were fixed with 4% PFA for 15 min at RT on the plates. No sucrose treatments or cryosections were performed on fixed hESCs. For immunostaining of both the aggregates and the hESCs, a 10% horse serum (Vector Laboratories) in 0.1% Triton X100 1× PBS solution was used for blocking, and a 3% horse serum in 0.1% Triton X100 1× PBS solution was used for primary and secondary antibody incubations. Primary antibodies used in this study were listed in Supplementary Table 1. All anti-CHD7 immunostaining assays in this study used the CHD7 antibody from R&D Systems (#AF7350). Alexa Fluor conjugated anti-mouse (IgG1, IgG2a, and IgG2b), rabbit, sheep, or goat secondary antibodies (Thermo Fisher) were used for primary antibody detection. ProLong Gold antifade reagent with DAPI (Thermo Fisher) was used to mount the samples and visualize cellular nuclei.

Microscopy images of hES cells and sectioned aggregates were captured on a Nikon A1R-HD25 confocal microscope or a Leica DMi8 inverted microscope.

### Western blot
hESCs were lysed in of RIPA buffer (Thermo Fisher) supplemented with 1× Halt protease inhibitor cocktail (Thermo Fisher) and 5 mM EDTA for 15 min on ice. After cell lysis, the sample were mixed with Laemmli sample buffer (Bio-Rad) and DTT (Thermo Fisher) to final concentrations of 1× and 25 mM, respectively. Samples were heated at 95 °C for 10 min, and then centrifuged at $14,000 \times g$ for 15 min to pellet the cell debris. Supernatants were loaded onto a 4–15% Mini-PROTEAN TGX precast gel (Bio-Rad) and were subject to electrophoresis in 1× running buffer (25 mM Tris, 192 mM glycine, 0.1% SDS, pH 8.3) (Bio-Rad) at 200 V for 35 min. After electrophoresis, samples were transferred to a PVDF membrane (Bio-Rad) using a Trans-Blot Turbo system (Bio-Rad) at 1.3 A, up to 25 V for 10 min. The PVDF membrane was briefly immersed in a wash buffer (0.05% Tween-20 (Sigma) in PBS), and then in a blocking buffer (0.05 g/mL blotting-grade blocker (Bio-Rad) and 0.05% Tween-20 in PBS) for 30 min shaking at RT. Primary antibodies and HRP-conjugated secondary antibodies were diluted in the blocking buffer, and antibody incubation was performed rocking overnight at 4 °C for primary antibodies and rocking for 1–2 h at RT for secondary antibodies. Three times of 10 min washing in the wash buffer was performed following primary and secondary antibody incubations. Bands were detected with an ECL substrate (Bio-Rad) and the membrane was imaged with a ChemiDoc imager (Bio-Rad). For HRP-conjugated primary antibodies, the secondary antibody incubation step and the subsequent washing steps were omitted. For blotting with a different primary antibody on the same membrane, stripping with a Restore Plus stripping buffer (Thermo Fisher) was performed to remove prior antibodies and chemiluminescent substrates. Primary antibodies used in this study are listed in Supplementary Table 1. Unless otherwise noted, all anti-CHD7 western blotting assays in this study used the CHD7 antibody from R&D Systems (#AF7350). Uncropped and unprocessed images of western blots and DNA gels are provided with the article.

### Statistics and reproducibility
For samples to be used for comparison of immunofluorescence intensities, all samples were processed for immunostaining at the same

time using the same tubes of diluted primary or secondary antibody mixtures, and samples were blocked, incubated, and washed for the same durations of time. These stained samples were imaged with the Nikon A1R-HD25 confocal microscope using the same image capture settings, including the same laser power, the same pinhole size, the same HV gain and offsets, etc. The exported TIFF images from each fluorescence channel were merged but were not adjusted in any other way prior to fluorescence intensity measurement in the Fiji software. The circle click tool from the ROI 1-click tool sets in Fiji was used to manually measure fluorescence intensities of nuclear proteins such as SOX2, HOXB9, and DLX5. The polygon selection tool from Fiji was used for whole-cell fluorescence intensity manual measurements for cell body-localized proteins COL9A2 and FBXO2. To visualize and locate the nuclei and the cell membranes of otic progenitors during measurement, $PAX2^{nG}$, and EPCAM channels were merged with the channel of interest. The Fiji software records intensity data from each channel separately, and only the data from the channel of interest were used for downstream analysis.

Graphical plots and statistical analysis of measured fluorescence intensity data were performed in GraphPad Prism 9. A total of 17,425 nuclei and cells from 129 total otic vesicles were measured and quantified (135.84 nuclei or cells from each otic vesicle on average). For each sample, data were obtained from three otic vesicles of three different aggregates from three independent experiments. All samples were subject to Anderson-Darling, D'Agostino & Pearson, Shapiro-Wilk, and Kolmogorov-Smirnov normality testing. Datasets containing sample(s) that did not pass the normality test were analyzed by Kruskal–Wallis test followed by Dunn's multiple comparisons test. Data collection and analysis were not performed blind to the conditions of the experiments. Violin plots display the full distribution of individual data points.

Prior to harvesting organoid samples for scRNA-seq or immunostaining, d19–d20 aggregates were pre-screened based on the epithelial $PAX2^{nG}$ fluorescence signals. d60–d70 WT and $CHD7^{KO/+}$ aggregates with the $POU4F3^{nT}$ reporter knockin were pre-screened based on the hair cell-specific $POU4F3^{nT}$ fluorescence signals. d60–d70 $CHD7^{KO/KO}$ aggregates were pre-screened based on the presence of vesicle structures when viewed under a phase-contrast microscope. Low-quality organoids with few $PAX2^{nG}$-positive epithelial vesicle structures, few $POU4F3^{nT}$-positive hair cells, or few vesicle structures were not used from subsequent scRNA-seq or immunostaining experiments.

All immunofluorescence, western blot, and DNA gel electrophoresis data shown in this article are representative of a minimum of three independent experiments with similar results.

### Reporting summary
Further information on research design is available in the Nature Portfolio Reporting Summary linked to this article.

## Data availability
The scRNA-seq data generated in this study have been deposited in the Gene Expression Omnibus with accession code GSE208585. This study used the GRCh38/hg38 human reference genome dataset (https://www.ncbi.nlm.nih.gov/assembly/GCF_000001405.26/). Source data generated in this study are provided with this paper. Source data are provided with this paper.

## Code availability
Scripts used for scRNA-seq analysis are available at [https://github.com/HashinoLab/Nie_et_al_CHD7] with https://doi.org/10.5281/zenodo.7139816[73].

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

## Acknowledgements

This work was supported by National Institute of Health grants R01DC015788 (E.H.), R01DC013294 (E.H.), and R21DC020160 (J.N.), the Indiana University Health-Indiana University School of Medicine Strategic Research Initiative (E.H.), and Indiana Clinical and Translational Sciences Institute core pilot grant (E.H.). The authors would like to thank J. Harkin, A. Botros, and S. Moore for their technical assistance. We thank the following Indiana University School of Medicine core facilities for their support: the Center for Medical Genomics, Flow Cytometry Resource Facility, and the Specimen Storage Facility.

## Author contributions

J.N. designed and led the study, performed experiments, analyzed data, and wrote the manuscript. Y.U. performed experiments and analyzed data. A.J.S. analyzed the data. E.H. conceived and designed the study, oversaw experiments, and wrote the manuscript. All authors read and approved the final manuscript.

## Competing interests

The authors declare no competing interests.
