## [Peer Review File · Nature Communications]

CHD7 regulates otic lineage specification and hair cell differentiation in human inner ear organoidsREVIEWER COMMENTS

Reviewer #1 (Remarks to the Author):

I have reviewed with great interest the manuscript from Nie and colleagues "CHD7 regulates otic lineage differentiation and deafness gene expression in human inner ear organoids" submitted to Nature Communications.

The lab of Dr Hashino has previously developed a human iPSC/ESC organoid model to derive otic progenitors and inner ear sensory and non-sensory components. Here the authors make use of this inner ear organoid model to study the effect of mutations in CHD7, a gene causative of CHARGE syndrome.

They develop and exploit reporter lines to label otic progenitors (PAX2-GFP reporters) and/or hair cells (POU4F3-Tomato), combine this with CRISPR editing to obtain mono or biallelic deletion/mutation of CHD7 and analyze the resulting phenotypes using histological characterization of the organoids and single cell RNA sequencing analysis.

CHD7 mutations result in significantly altered expression of otic genes at early developmental stages (d20 in vitro), eventually resulting in defects in sensory epithelia differentiation (day 70 in vitro). This provides a potential explanation for the hearing deficit observed in CHARGE patients.

The study is the first of its kind to make use of inner ear organoids to address in such detail the molecular basis of genetic mutations associated to hearing loss in human. The experimental methodology is sound and uses state-of-the-art approaches for characterization of the studied mutation. I am very excited about this significant progress for the field.

The manuscript is well written, clear to follow and I am sure it will be of great interest for the scientific community and I'd like to congratulate the team for their work.

I have a couple of comments/revisions/suggestion for the authors for their manuscript.

Comments needing revisions

1.1 In Figure 2 the differentiation of WT and mutant lines towards otic progenitors (panels d-t) and sensory epithelia (panels e-w) is analyzed. I am puzzled by the remaining SOX2 expression at day 70 of differentiation. The authors suggest that supporting cells are completely absent. The remaining SOX2 cells in panel r,v,s,w could be supporting cells (though lacking SPARCL1 expression) if what the panels show is otic tissue. If not, what are they? Are the authors looking at prosensory epithelia (SOX2+) that fail to differentiate hair cells? or are these regions SOX2+ neural tissue/neural rosette? Or other type of tissue (see below 1.2).

With the high magnification panels provided it is difficult to judge. Representative images of whole organoid histology could be provided to illustrate the overall changes and complete absence of SC/HC. In addition, more markers could be used to characterize these cell populations and give a better idea of the resulting phenotype of KO organoids (see also point 1.3).

1.2 in figure 4 top panel. scRNA-seq shows complete absence of SOX2 in PAX2-GFP progenitors but SOX2 expression remains very high in the neighboring tissue (IF figure 4a) What kind of tissue is this? Are these the same cells as above (comment 1.1) ? SOX2+ but not otic/non epithelial? Is it population "1" in UMAP figure S7b?

TUBB3 positive? This could be discussed more in detail.

1.3 The authors further study the changes in gene expression by scRNAseq in WT and CHD7 KO/+ hair cells and supporting cells at day 70 of differentiation (Fig 5), which morphologically seem undistinguishable from their WT counterpart (Figure 2). Several changes are observed, in particular in the hair cell population, including downregulation of deafness genes. Supporting cells seem less affected (similar UMAPs between the two cell lines). Why have the authors not looked at RNAseq of organoids derived from CHD7 KO/KO lines (whole organoids without sorting). I am surprised the data

is not included. This could be used to illustrate the lack of hair cells and supporting cells, as well as clarify what becomes of the "odd" otic progenitor populations described at day 20 in the KO/KO mutant! This could be a rather straightforward way to characterize the consequences of CHD7 KO at the level of sensory epithelia. I would recommend including the data set.

Suggestion

1) Figure 3 and supplementary figure 7 show the scRNAseq of day 20 PAX2-GFP sorted cells. This is a very interesting dataset as it shows the global changes occurring in this cell population. I would suggest changing the labeling in "panel A main figure 3" and removing the text ("mainly wt"; "mainly CHD7 mutant") and leaving instead only the population names. panel B (main figure 3), could be replaced with the UMAP plots of supplementary figure 7 panel C. This would make the figure more clear in my opinion (without need to go back and forward with supplement) and illustrate better the global changes in gene expression resulting from CHD7 KO, in particular also the appearance of additional cell populations.

Reviewer #2 (Remarks to the Author):

The authors reported the change of gene expression patterns in inner ear organoids induced from human ESCs with ablation of CHD7 alleles or ATP-dependent activity. They suggested the candidate of CHD7 downstream related to the inner ear development. By comparing the comprehensive expression data from CHD7 KO and mutant cell lines, the authors found that the dysregulation of some genes was not dependent on the ATPase activity of CHD7. Moreover, the mixture of the wild-type (WT) and knockout (KO) cells resulted in the expression recovery of some dysregulated genes in KO cells. This study was based on many well-designed experiments, and the results seemed to be robust. The results of this study are intriguing for the clinical treatment of CHARGE syndrome. However, several points should be clarified, and some discussion should be revised.

Major points

1. The organoids induced from human ESCs contained both hair and supporting cells. However, the authors did not specify if they were cochlear or vestibular cells. The authors will be able to confirm whether either or both cochlear and vestibular cells were induced in the organoids by analyzing their RNAseq data. By clarifying the characteristics of the hair and supporting cells, the significance of this study will become more apparent. Moreover, the "deafness genes," including TBX1, LMX1A, and SOX10, which the authors described in the manuscript, affect the development of both vestibular organs and cochleae. Therefore, if the authors cannot determine the characteristics of the hair and supporting cells in the organoids, they should change the naming of the set of these genes.
2. In the discussion, the authors described the down-regulated expression of SOX2 will affect the malformation of the cochlea or semicircular canals because of the malformed inner ears of the Sox2 mutant mouse. However, the most popular inner ear malformations in CHARGE syndrome are semicircular-canal hypoplasia or aplasia and cochlear hypoplasia, but not truncated semicircular canals observed in Sox2 mutant mice. Moreover, the interaction between epithelial tissue and mesenchyme forms the inner ear. Therefore, the authors should be careful to conclude that SOX2, which is expressed in ectodermal tissue, correlates with the inner ear malformation observed in CHARGE syndrome.
3. Most patients with CHARGE syndrome have a heterozygous mutation in CHD7, although this study analyzed homozygous KO and mutant mice as well as heterozygous mice. And the authors found that the dysregulation of some genes was observed only in the homozygous mutant (Sox10), and other

genes showed dysregulation similarly in heterozygous and homozygous mutation (COL9A2). The authors should emphasize the scientific significance of the analysis of homozygous mutation

Minor points

1. In CHARGE syndrome patients, a hypoplastic cochlear nerve is also observed frequently and affects the outcome of cochlear implants. Therefore, any data related to neuronal cells within organoids will help understand and treat CHARGE syndrome.

Reviewer #3 (Remarks to the Author):

This manuscript describes the role of CHARGE syndrome gene CHD7 in the conversion of iPS cells to sensory hair cells. The authors use human iPS cells to look at the differentiation of otic progenitors and the conversion of progenitors to the cells of inner ear sensory epithelia. They do heterozygous and homozygous knockout of CHD7 and in addition mutate S834 of CHD7, an important residue for its ATPase activity, to inactivate the chromatin remodeling function of CHD7. They perform transcriptional analysis to determine changes due to chromatin remodeling and to non-chromatin effects of CHD7. They show that the transcriptional activity downstream of the S834F mutation differs from that found after knockout, leading to the conclusion that some of CHD7 activities are not mediated by chromatin remodeling. The authors find a complete lack of supporting cell and hair cell formation after homozygous KO as well as homozygous point mutation.

These are remarkable findings of broad significance both for the understanding of the potential roles for CHD7 and of the syndrome's effects on the inner ear. The results are well documented and although not confirmed in vivo, they have uncovered new roles of CHD7 in human cells and have demonstrated relevance of these effects to CHARGE syndrome by introducing disease mutations into human inner ear cells.

However several issues should be addressed:

The authors reference the previous work well but do not fully acknowledge some of the relevant results in those papers. Others have found that hair cells developed normally in heterozygous CHD7 KO mice (Adams et al, 2007); that CHD7 controlled by let7 miRNA affected hair cell progenitors (Evsen et al, 2020); that hair cells developed normally but degenerated with noise exposure of the pups (Ahmed et al, 2021). An additional set of papers showing a concentration dependent effect of SOX2 (Neves et al, 2012; Kempfle et al, 2015) support their hypothesis that alterations in SOX2 expression could account for the hair cell and supporting cell differences observed in the CHD7 KO.

While the advantages of using human iPS cells are explained and reasonable, there should be qualifiers regarding limitations of the in vitro approach - the difficulty in monitoring cellular interactions and proximity-related cues, and the lack of insight into hair cell activity, sensitivity of the cells to stress, and effects of CHD7 on morphogenesis of middle ear and semicircular canals. Thus for example, vestibular problems were attributed to defects in the formation of the semicircular canals (Adams et al, 2007). They point out that embryonic lethality of CHD7 homozygous mutants is a problem, but should also point out that the results with heterozygous mutants are valuable because of the prevalence of heterozygosity in patients with CHARGE syndrome.

Extended data on immunostaining was mentioned in the text but not included with the materials. These experimental details are needed for the assessment of changes in immunostaining.

Dear Reviewers:

Thank you very much for your interest in our work as well as a prompt and thorough review of our manuscript. In response to your suggestions, we have modified the manuscript as outlined below (modified and added texts shown in blue color in the manuscript) and added 1 additional main figure, 9 additional supplementary figures, and 2 additional supplementary tables. We believe that these modifications greatly strengthen the clarity of the manuscript. We apologize for the long time needed for the revision, as some of the experiments, especially the day 70 scRNA-seq, requires extended time for sample growth and data generation/analysis. We very much appreciate your review of this revised manuscript.

Reviewer #1 (Remarks to the Author):

General Comments: *I have reviewed with great interest the manuscript from Nie and colleagues “CHD7 regulates otic lineage differentiation and deafness gene expression in human inner ear organoids” submitted to Nature Communications.*

The lab of Dr Hashino has previously developed a human iPSC/ESC organoid model to derive otic progenitors and inner ear sensory and non-sensory components. Here the authors make use of this inner ear organoid model to study the effect of mutations in CHD7, a gene causative of CHARGE syndrome. They develop and exploit reporter lines to label otic progenitors (PAX2-GFP reporters) and/or hair cells (POU4F3-Tomato), combine this with CRISPR editing to obtain mono or biallelic deletion/mutation of CHD7 and analyze the resulting phenotypes using histological characterization of the organoids and single cell RNA sequencing analysis. CHD7 mutations result in significantly altered expression of otic genes at early developmental stages (d20 in vitro), eventually resulting in defects in sensory epithelia differentiation (day 70 in vitro). This provides a potential explanation for the hearing deficit observed in CHARGE patients.

The study is the first of its kind to make use of inner ear organoids to address in such detail the molecular basis of genetic mutations associated to hearing loss in human. The experimental methodology is sound and uses state-of-the-art approaches for characterization of the studied mutation. I am very excited about this significant progress for the field.

The manuscript is well written, clear to follow and I am sure it will be of great interest for the scientific community and I'd like to congratulate the team for their work.

Response: Thank you very much for your unequivocal endorsement of our manuscript.

Comment 1: *I have a couple of comments/revisions/suggestion for the authors for their manuscript. Comments needing revisions: In Figure 2 the differentiation of WT and mutant lines towards otic progenitors (panels d-t) and sensory epithelia (panels e-w) is analyzed. I am puzzled by the remaining SOX2 expression at day 70 of differentiation. The authors suggest that supporting cells are completely*

absent. The remaining SOX2 cells in panel r,v,s,w could be supporting cells (though lacking SPARCL1 expression) if what the panels show is otic tissue. If not, what are they? Are the authors looking at prosensory epithelia (SOX2+) that fail to differentiate hair cells? or are these regions SOX2+ neural tissue/neural rosette? Or other type of tissue (see below 1.2).

With the high magnification panels provided it is difficult to judge. Representative images of whole organoid histology could be provided to illustrate the overall changes and complete absence of SC/HC. In addition, more markers could be used to characterize these cell populations and give a better idea of the resulting phenotype of KO organoids (see also point 1.3).

Response: To investigate the cell identity of SOX2+ cells in d70 *CHD7*^{KO/KO} organoids, we performed d70 scRNA-seq on WT and *CHD7*^{KO/KO} organoid cells (Fig. 6 and Supplementary Figs. 14–17). Our results showed that these cells express otic lineage genes including *FBXO2*, *OC90*, *S100A1*, *CLDN6*, and *PLEKHB1*. However, key supporting cell markers¹⁻⁴ *SPARCL1*, *BRICD5*, and *OTOG* are absent in these cells. Instead, genes in the keratinocyte/epidermis development gene ontology (GO) categories were upregulated (Fig. 2u and Supplementary Figs. 16–17). These results suggest that these SOX2+ cells are not supporting cells, but rather they represent otic-like tissues with drifted cellular identity towards the epidermis lineage.

In addition, we have compared these otic-like mutant cells with WT supporting cells, and showed differentially expressed genes (downregulation of otic genes such as *OTOL1*, *SPARCL1*, and *COL9A2*, upregulation of keratinocyte genes such as *KRT7*, *KRT6B*, and *KRT17*) and gene sets (e.g. inner ear morphogenesis and anchoring junction GO categories) (Supplementary Figs. 17d–f).

We have added low magnification images showing the whole organoid histology (Supplementary Fig. 15). Analysis with additional marker genes was shown in Supplementary Figs. 14, 16 and 17.

Comment 2: *In figure 4 top panel, scRNA-seq shows complete absence of SOX2 in PAX2-GFP progenitors but SOX2 expression remains very high in the neighboring tissue (IF figure 4f) What kind of tissue is this? Are these the same cells as above (comment 1.1)? SOX2+ but not otic/non-epithelial? Is it population “1” in UMAP figure S7b? TUBB3 positive? This could be discussed more in detail.*

Response: To investigate the cell identity of SOX2+ cells outside of otic vesicles, we performed scRNA-seq with the *PAX2*^{nG+} and *PAX2*^{nG-} populations of d20 WT organoids (Supplementary Fig. 8). scRNA-seq and immunofluorescence marker gene analysis suggests these SOX2+ cells in WT and *CHD7*^{KO/KO} mutant organoids are *S100B*⁺ *SOX10*⁺ *FOXD3*⁺ *PAX3*⁺ neural crest cells (Supplementary Fig. 9).

Comment 3: *The authors further study the changes in gene expression by scRNA-seq in WT and CHD7 KO/+ hair cells and supporting cells at day 70 of differentiation (Fig 5), which morphologically seem undistinguishable from their WT counterpart (Figure 2). Several changes are observed, in particular in the hair cell population, including downregulation of deafness genes. Supporting cells seem less affected*

(similar UMAPs between the two cell lines). Why have the authors not looked at scRNA-seq of organoids derived from CHD7 KO/KO lines (whole organoids without sorting). I am surprised the data is not included. This could be used to illustrate the lack of hair cells and supporting cells, as well as clarify what becomes of the “odd” otic progenitor populations described at day 20 in the KO/KO mutant! This could be a rather straightforward way to characterize the consequences of CHD7 KO at the level of sensory epithelia. I would recommend including the data set.

Response: In response to your comments, we performed d70 scRNA-seq with WT and *CHD7*^{KO/KO} organoid cells (Fig. 6 and Supplementary Figs. 14–17). Cluster annotation analysis revealed that the merged dataset contains a single hair cell cluster composed solely of WT cells ($n = 2,810$) without any *CHD7*^{KO/KO} cells ($n = 0$) (Fig. 6), confirming the failure of hair cell generation in *CHD7*^{KO/KO} organoids (Fig. 2u–w). As mentioned above (Response to Comment 1), the SOX2⁺ cells in the d70 *CHD7*^{KO/KO} vesicle structures appear to be otic-like cells, but are not supporting cells. Based on otic gene expression, it appears that the “odd” otic progenitor in d20 *CHD7*^{KO/KO} organoids have become these SOX2⁺ ANXA4^{low} otic-like cells, as well as an ANXA4^{high} ATOH1⁻ EPCAM⁺ FBXO2^{low} OC90^{low} S100A1^{low} CLDN6^{low} SOX2^{low} population with unknown cell identity (Cluster 3 in Fig. 6) (Supplementary Fig. 16).

Comment 4: *Suggestion: Figure 3 and supplementary figure 7 show the scRNA-seq of day 20 PAX2-GFP sorted cells. This is a very interesting dataset as it shows the global changes occurring in this cell population. I would suggest changing the labeling in “panel A main figure 3” and removing the text (“mainly wt”; “mainly CHD7 mutant”) and leaving instead only the population names. panel B (main figure 3), could be replaced with the UMAP plots of supplementary figure 7 panel C. This would make the figure more clear in my opinion (without need to go back and forward with supplement) and illustrate better the global changes in gene expression resulting from CHD7 KO, in particular also the appearance of additional cell populations.*

Response: Based on the reviewer’s suggestion, Fig. 3b has been replaced with an UMAP plot showing all cells instead of only the otic progenitors. The labeling in Fig. 3a has been updated. However, the genotype labeling (“mainly WT”; “mainly CHD7 mutant”) for the otic progenitor clusters were kept in place in Fig. 3a as we feel it could make the figure easier to follow.

Reviewer #2 (Remarks to the Author):

General Comments: *The authors reported the change of gene expression patterns in inner ear organoids induced from human ESCs with ablation of CHD7 alleles or ATP-dependent activity. They suggested the candidate of CHD7 downstream related to the inner ear development. By comparing the comprehensive expression data from CHD7 KO and mutant cell lines, the authors found that the dysregulation of some genes was not dependent on the ATPase activity of CHD7. Moreover, the mixture of the wild-type (WT) and knockout (KO) cells resulted in the expression recovery of some dysregulated genes in KO cells. This*

study was based on many well-designed experiments, and the results seemed to be robust. The results of this study are intriguing for the clinical treatment of CHARGE syndrome. However, several points should be clarified, and some discussion should be revised.

Response: Thank you for your positive comments on our manuscript and constructive and insightful reviews.

Comment 1: *Major points: The organoids induced from human ESCs contained both hair and supporting cells. However, the authors did not specify if they were cochlear or vestibular cells. The authors will be able to confirm whether either or both cochlear and vestibular cells were induced in the organoids by analyzing their scRNA-seq data. By clarifying the characteristics of the hair and supporting cells, the significance of this study will become more apparent. Moreover, the “deafness genes,” including TBX1, LMX1A, and SOX10, which the authors described in the manuscript, affect the development of both vestibular organs and cochleae. Therefore, if the authors cannot determine the characteristics of the hair and supporting cells in the organoids, they should change the naming of the set of these genes.*

Response: We have provided additional data and discussion regarding the vestibular hair cell identity. Our previous studies suggested that the hair cells generated in our current version of human inner ear organoids exhibit electrophysiological, morphological, and marker gene expression characteristics reminiscent of vestibular hair cells⁵⁻⁷. The vestibular/cochlear marker gene analysis in our current study has confirmed these findings (Supplementary Figs. 12).

We still believe that keeping the “deafness genes” naming is the best way to describe the dysregulated hearing loss-related genes in this study for the following reasons:

- a. These genes are from the OtoSCOPE deafness gene panel⁸, which curates genes involved in otic lineage cell functions (e.g. mechanosensory function of hair cells) identified in patients with profound hearing loss. A comprehensive panel for vestibular dysfunction genes is not available, as the genetic causes of vestibular disorders are less well studied than hearing loss⁹, leading to likely underreported vestibular dysfunction genes and the lack of vestibular gene panel curating.
- b. *CHD7* and all the *CHD7* regulated deafness genes identified in this study are expressed in both the cochlea and the vestibule during mouse development (Supplementary Fig. 20. Data from Scheffer et al. 2015)^{3,10}. These data provide the gene expression basis for their potentially shared function in both the cochlea and the vestibule.
- c. 67% of the deafness genes identified in this study are also known to be associated with vestibular dysfunctions and defects (Supplementary Table 2), despite the fact that the genetic causes of vestibular disorders are less well studied than hearing loss, leading to likely underreported vestibular dysfunction genes and the lack of comprehensive vestibular dysfunction gene panels⁹. These data suggest that these genes likely play similar roles related to mechanical sensing both in the cochlea and in the vestibule.
- d. Our current study provides the first transcriptome-wide analysis of *CHD7* mutants otic lineage cells. The dysregulated deafness genes identified in this vestibular system could serve as good

candidate genes for future studies in cochlear *in vitro* and *in vivo* CHARGE syndrome disease models.

These discussions are included in lines #335 - #347 of the Discussion section.

Comment 2: *In the discussion, the authors described the down-regulated expression of SOX2 will affect the malformation of the cochlea or semicircular canals because of the malformed inner ears of the Sox2 mutant mouse. However, the most popular inner ear malformations in CHARGE syndrome are semicircular-canal hypoplasia or aplasia and cochlear hypoplasia, but not truncated semicircular canals observed in Sox2 mutant mice. Moreover, the interaction between epithelial tissue and mesenchyme forms the inner ear. Therefore, the authors should be careful to conclude that SOX2, which is expressed in ectodermal tissue, correlates with the inner ear malformation observed in CHARGE syndrome.*

Response: Based on the reviewer's comment, the discussion regarding SOX2 has been modified in lines #308 – 325 of the Discussion section. We believe that combined effects of many *CHD7* downstream genes contribute to the CHARGE syndrome phenotype, but SOX2 might be one of the most important ones.

With regard to the interaction between epithelial tissue and mesenchyme, Balendran et al. (2021) suggested that *Chd7* expression in the otic mesenchyme is dispensable for inner ear morphogenesis, as conditional knockout of *Chd7* in the mouse otic mesenchyme resulted in normal vestibular and cochlear morphologies¹¹. These results strongly suggest that *Chd7* downstream genes in the otic epithelial tissues play more important roles in CHARGE syndrome phenotypes. Therefore, we feel like the fact that SOX2 has low expression levels in the mesenchymal tissues (Supplementary Fig. 8) does not disqualify SOX2 as one of the candidate genes responsible for CHARGE phenotypes.

Comment 3: *Most patients with CHARGE syndrome have a heterozygous mutation in CHD7, although this study analyzed homozygous KO and mutant organoids as well as heterozygous organoids. And the authors found that the dysregulation of some genes was observed only in the homozygous mutant (Sox10), and other genes showed dysregulation similarly in heterozygous and homozygous mutation (COL9A2). The authors should emphasize the scientific significance of the analysis of homozygous mutation.*

Response: The discussion regarding the scientific significance of the analysis of homozygous mutants has been included in lines #296 - #300 of the Discussion section. The homozygous mutations should generally lead to more severe/obvious dysregulation of downstream genes. Therefore, analysis of the homozygotes should provide a more comprehensive list of downstream genes. In addition, analysis of the homozygotes reveals the true developmental significance of *CHD7*. For example, reduced *CHD7* expression in the heterozygotes still supports the generation of hair cells and supporting cells with normal morphologies. However, analysis of the homozygous mutants revealed that hair cells and

supporting cells are completely absent without *CHD7*, suggesting a critical role of *CHD7* in hair cell and supporting cell differentiation.

Comment 4: *Minor points. In CHARGE syndrome patients, a hypoplastic cochlear nerve is also observed frequently and affects the outcome of cochlear implants. Therefore, any data related to neuronal cells within organoids will help understand and treat CHARGE syndrome.*

Response: scRNA-seq and immunofluorescence analyses of the neuronal cells which innervate hair cells in organoids are now included (Supplementary Fig. 13).

In *CHD7*^{KO/+} organoids, the neuronal population showed minimal gene expression disruptions, with only a few ribosomal genes being downregulated (Supplementary Fig. 13a–e). Immunostaining of the neurons revealed normal neurite infiltration to the sensory epithelium and normal contact with hair cells (Supplementary Fig. 13f–i), which is consistent with the normal innervation pattern found in the utricle and saccule of *Chd7* heterozygous deletion mice¹². These data suggests that the sensory neurons appear largely unaffected by the mono-allelic loss of *CHD7*.

In the *CHD7*^{KO/KO} organoids, occasional NEFL⁺ neurite infiltration can be observed in these otic-like vesicle structures, albeit at a much lower rate compared to neuron innervation in the WT and the *CHD7*^{KO/+} sensory epithelium (Supplementary Fig. 13j–k).

Reviewer #3 (Remarks to the Author):

General Comments: *This manuscript describes the role of CHARGE syndrome gene CHD7 in the conversion of iPS cells to sensory hair cells. The authors use human iPS cells to look at the differentiation of otic progenitors and the conversion of progenitors to the cells of inner ear sensory epithelia. They do heterozygous and homozygous knockout of CHD7 and in addition mutate S834 of CHD7, an important residue for its ATPase activity, to inactivate the chromatin remodeling function of CHD7. They perform transcriptional analysis to determine changes due to chromatin remodeling and to non-chromatin effects of CHD7. They show that the transcriptional activity downstream of the S834F mutation differs from that found after knockout, leading to the conclusion that some of CHD7 activities are not mediated by chromatin remodeling. The authors find a complete lack of supporting cell and hair cell formation after homozygous KO as well as homozygous point mutation.*

These are remarkable findings of broad significance both for the understanding of the potential roles for CHD7 and of the syndrome's effects on the inner ear. The results are well documented and although not confirmed in vivo, they have uncovered new roles of CHD7 in human cells and have demonstrated relevance of these effects to CHARGE syndrome by introducing disease mutations into human inner ear cells.

Response: Thank you for your thorough and constructive review of our manuscript and your positive comments.

Comment 1: *However, several issues should be addressed: The authors reference the previous work well but do not fully acknowledge some of the relevant results in those papers. Others have found that hair cells developed normally in heterozygous CHD7 KO mice (Adams et al, 2007); that CHD7 controlled by let7 miRNA affected hair cell progenitors (Evsen et al, 2020); that hair cells developed normally but degenerated with noise exposure of the pups (Ahmed et al, 2021). An additional set of papers showing a concentration dependent effect of SOX2 (Neves et al, 2012; Kempfle et al, 2015) support their hypothesis that alterations in SOX2 expression could account for the hair cell and supporting cell differences observed in the CHD7 KO.*

Response: Based on the reviewer's comment, we have included the references of previous works in lines #42 – 52 in the Introduction section (CHD7 references) and in lines #312 – 313 of the Discussion section (SOX2 references).

Comment 2: *While the advantages of using human iPS cells are explained and reasonable, there should be qualifiers regarding limitations of the in vitro approach - the difficulty in monitoring cellular interactions and proximity-related cues, and the lack of insight into hair cell activity, sensitivity of the cells to stress, and effects of CHD7 on morphogenesis of middle ear and semicircular canals. Thus for example, vestibular problems were attributed to defects in the formation of the semicircular canals (Adams et al, 2007). They point out that embryonic lethality of CHD7 homozygous mutants is a problem, but should also point out that the results with heterozygous mutants are valuable because of the prevalence of heterozygosity in patients with CHARGE syndrome.*

Response: We have now included a paragraph discussing the limitations of the inner ear organoid model system (lines #381 – 391). Additionally, the discussion regarding heterozygous and homozygous mutants has been updated in lines #296 – 300 of the Discussion section.

Comment 3: *Extended data on immunostaining was mentioned in the text but not included with the materials. These experimental details are needed for the assessment of changes in immunostaining.*

Response: We apologize that the supplementary table detailing the immunostaining and western blot antibody information was mistakenly omitted when uploading our manuscript. The information is now included as Extended Data Table 1.

References:

1. Burns, J.C., Kelly, M.C., Hoa, M., Morell, R.J. & Kelley, M.W. Single-cell RNA-Seq resolves cellular complexity in sensory organs from the neonatal inner ear. *Nat Commun* **6**, 8557 (2015).

2. Wilkerson, B.A. *et al.* Novel cell types and developmental lineages revealed by single-cell RNA-seq analysis of the mouse crista ampullaris. *Elife* **10** (2021).
3. Scheffer, D.I., Shen, J., Corey, D.P. & Chen, Z.Y. Gene Expression by Mouse Inner Ear Hair Cells during Development. *J Neurosci* **35**, 6366-6380 (2015).
4. Jan, T.A. *et al.* Spatiotemporal dynamics of inner ear sensory and non-sensory cells revealed by single-cell transcriptomics. *Cell Rep* **36**, 109358 (2021).
5. Koehler, K.R. *et al.* Generation of inner ear organoids containing functional hair cells from human pluripotent stem cells. *Nat Biotechnol* **35**, 583-589 (2017).
6. Koehler, K.R., Mikosz, A.M., Molosh, A.I., Patel, D. & Hashino, E. Generation of inner ear sensory epithelia from pluripotent stem cells in 3D culture. *Nature* **500**, 217-221 (2013).
7. Liu, X.P., Koehler, K.R., Mikosz, A.M., Hashino, E. & Holt, J.R. Functional development of mechanosensitive hair cells in stem cell-derived organoids parallels native vestibular hair cells. *Nat Commun* **7**, 11508 (2016).
8. Shearer, A.E. *et al.* Comprehensive genetic testing for hereditary hearing loss using massively parallel sequencing. *Proc Natl Acad Sci U S A* **107**, 21104-21109 (2010).
9. Mei, C. *et al.* Genetics and the Individualized Therapy of Vestibular Disorders. *Front Neurol* **12**, 633207 (2021).
10. Hurd, E.A., Poucher, H.K., Cheng, K., Raphael, Y. & Martin, D.M. The ATP-dependent chromatin remodeling enzyme CHD7 regulates pro-neural gene expression and neurogenesis in the inner ear. *Development* **137**, 3139-3150 (2010).
11. Balendran, V. *et al.* Chromatin remodeler CHD7 is critical for cochlear morphogenesis and neurosensory patterning. *Dev Biol* **477**, 11-21 (2021).
12. Adams, M.E. *et al.* Defects in vestibular sensory epithelia and innervation in mice with loss of Chd7 function: implications for human CHARGE syndrome. *J Comp Neurol* **504**, 519-532 (2007).

REVIEWER COMMENTS

Reviewer #1 (Remarks to the Author):

I have reviewed the revised version of the manuscript "CHD7 regulates otic lineage specification and hair cell differentiation in human inner ear organoids" from the lab of Dr Hashino. The work is very impressive and the new data supports very strongly the claim that CHD7 mutation(s) affect otic lineage differentiation and eventually result in the absence of hair cells in this model. The data is presented very clearly and the images are truly beautiful. The new data on cochlear vs vestibular hair cell markers in the d70 organoids, as well as on neuronal innervation, are also very interesting and will be useful for the community. I have two final minor comments.

1)

In the previous revision I asked what kind of cells were the SOX2+ cells present in the S824F/S824F and KO/KO mutant outside the vesicle at day 20 (Fig 4) and forming the vesicle by day 70 (Fig1). The authors provide now scRNAseq data of d20 WT organoids (PAX2 GFP+ and PAX2 GFP-) and indicate SOX2+/GFP- cells are likely neural crest (NC) cells (Suppl 8 & 9). This is the case for the WT cells only. It remains unclear what they are for the mutant lines (NOT SOX10+, also Fig 4). In supplementary figure 16, the new sequencing data from the KO/KO line at day 70 show that SOX2+ cells are "odd" otic-like cells which have taken up epidermal-like signature. Very interesting! As globally gene expression is altered for the mutant lines (homozygotes) it may be hard to establish exactly which cell type one is looking at. Maybe the authors could add a comment in the text (line 190) concerning this. Now it reads as SOX2+ cells in KO/KO cells outside of the vesicle are NC cells.

2)

The authors display the quantification of the immunofluorescence (wt vs mutants (Fig 4) and single vs chimeric organoids (Fig7)) using violin plots, with median and quartile value indicated. While revising the manuscript I noticed that in the figure legends they indicate that: "3 otic vesicles (OV) per genotype have been quantified". My question is 3 OV for how many organoids? and how many independent experiments? I was not able to find this in the method section. Is the data sets only consisting of n=3 vesicles? Then the violin plots are misleading and I believe not correctly used. More data point/independent repeats would be needed, in particular for the chimeric organoids experiments. The authors should clarify and possibly display the data points.

I am looking forward to see the finalized manuscript.

Reviewer #2 (Remarks to the Author):

The authors responded well to the issues raised by the reviewer. However, the revised manuscript revealed that hair and supporting cells induced in this study were similar to vestibular cells. This suggests that the results presented in this research do not always describe the function of CHD7 in the cochlea. For example, the authors showed that vestibular supporting cells were less affected by the mono-allelic loss of CHD7 (lines 208-209). However, we do not know if this result can be applied to the cochlear supporting cells. Therefore, the authors should stress this in the discussion, although they discussed the similarity of cochlear and vestibular systems.

Reviewer #3 (Remarks to the Author):

Revised manuscript is suitable for publication.

Dear Reviewers,

Thank you very much for your thorough review of our revised manuscript! In response to your suggestions, we have modified the manuscript as outlined below (modified texts shown in blue color in the manuscript). We very much appreciate your review of the second revision of the manuscript.

Reviewer #1 (Remarks to the Author):

General Comments: *I have reviewed the revised version of the manuscript "CHD7 regulates otic lineage specification and hair cell differentiation in human inner ear organoids" from the lab of Dr Hashino. The work is very impressive and the new data supports very strongly the claim that CHD7 mutation(s) affect otic lineage differentiation and eventually result in the absence of hair cells in this model. The data is presented very clearly and the images are truly beautiful.*

The new data on cochlear vs vestibular hair cell markers in the d70 organoids, as well as on neuronal innervation, are also very interesting and will be useful for the community. I have two final minor comments.

Response: Thank you very much for your detailed review and positive comments of our revised manuscript!

Comment 1: *In the previous revision I asked what kind of cells were the SOX2+ cells present in the S824F/S824F and KO/KO mutant outside the vesicle at day 20 (Fig 4) and forming the vesicle by day 70 (Fig2). The authors provide now scRNAseq data of d20 WT organoids (PAX2 GFP+ and PAX2 GFP-) and indicate SOX2+/GFP- cells are likely neural crest (NC) cells (Suppl 8 & 9). This is the case for the WT cells only. It remains unclear what they are for the mutant lines (NOT SOX10+, also Fig 4).*

In supplementary figure 16, the new sequencing data from the KO/KO line at day 70 show that SOX2+ cells are "odd" otic-like cells which have taken up epidermal-like signature. Very interesting!

As globally gene expression is altered for the mutant lines (homozygotes) it may be hard to establish exactly which cell type one is looking at. Maybe the authors could add a comment in the text (line 190) concerning this. Now it reads as SOX2+ cells in KO/KO cells outside of the vesicle are NC cells.

Response: Our scRNA-seq evidence for the neural crest cell identity is indeed for WT organoids

only (Supplementary Figs. 8, 9a–d). However, In Supplementary Fig. 9e–f, we have provided immunofluorescence evidence that not only in WT organoids, but also in *CHD7*^{KO/KO} organoids, the SOX2+ cells outside the otic vesicles appear to be S100B+ neural crest cells. S100B is a highly specific marker for neural crest cells in d20 WT inner ear organoids, as it's only found in the neural crest cluster (Supplementary Fig. 8f). Other neural crest markers such as SOX2, SOX10, ERBB3, FOXD3, PAX3, and MPZ are also expressed in this cluster (Supplementary Fig. 9d).

It should be noted that Figs. 4an–ao did not prove the SOX2+ cells outside otic vesicles in the mutant lines are SOX10-negative. First, SOX10+ peri-otic vesicle cells are present in *CHD7*^{S834F/S834F} organoids (Fig. 4an, note the red colored cells in the lower panel. In the upper panel the signal is partially blocked by the figure labeling). In addition, the fact that SOX10 signal is not observed in Fig. 4ao does not mean the SOX2+ peri-otic vesicle cells in *CHD7*^{KO/KO} organoids are SOX10-. Those SOX2+ cells do not always surround all otic vesicles, therefore those SOX2+ cells may or may not be present in a high magnification otic vesicle image.

To improve the clarity of the manuscript, we slightly revised the manuscript text at lines #190–192.

Comment 2: *The authors display the quantification of the immunofluorescence (wt vs mutants (Fig 4) and single vs chimeric organoids (Fig7)) using violin plots, with median and quartile value indicated.*

While revising the manuscript I noticed that in the figure legends they indicate that: " 3 otic vesicles (OV) per genotype have been quantified".

My question is 3 OV for how many organoids? and how many independent experiments? I was not able to find this in the method section. Is the data sets only consisting of n=3 vesicles? Then the violin plots are misleading and I believe not correctly used. More data point/independent repeats would be needed, in particular for the chimeric organoids experiments. The authors should clarify and possibly display the data points.

I am looking forward to see the finalized manuscript.

Response: As explained below, each sample in the violin plots contains ~400 data points (nuclei and/or cells), not 3 data points (otic vesicles).

For each otic vesicle (OV), we individually measured the fluorescence intensity of each nucleus (for nuclear proteins such as SOX2 and SIX1) or cell (for cytoplasmic proteins such as COL9A2 and FBXO2) and used these data points for quantification. On average, 135.84 (136.87 for Figure 4 and 131.38 for Figure 7) data points (nuclei or cells) were measured and quantified for each otic vesicle. We used 3 otic vesicles (OVs) from three different aggregates from three

independent experiments, therefore, each sample contains an average of 135.84 nuclei/cells × 3 OV_s = 407.52 data points. The sample size and quantification information are now clarified in Figure 4 and Figure 7 legends (lines #669–675, #719–721) as well as in the Methods section (lines #1293–1296).

Since each sample contains far more than 10 data points (An average of 407.52 data points per sample), it is not feasible to display all individual data points in a plot/graph. Rather, we took advantage of the density curve feature of violin plots, while also included median and quartile information to depict data distribution.

Reviewer #2 (Remarks to the Author):

Comment 1: *The authors responded well to the issues raised by the reviewer. However, the revised manuscript revealed that hair and supporting cells induced in this study were similar to vestibular cells. This suggests that the results presented in this research do not always describe the function of CHD7 in the cochlea. For example, the authors showed that vestibular supporting cells were less affected by the mono-allelic loss of CHD7 (lines 208-209). However, we do not know if this result can be applied to the cochlear supporting cells. Therefore, the authors should stress this in the discussion, although they discussed the similarity of cochlear and vestibular systems.*

Response: In response to your comments, we now stressed that the lack of cochlear hair cells is a major limitation of the current human inner ear organoid model, and added that “... the functions of *CHD7* in cochlear hair cells and supporting cells cannot be directly inferred from this study” in the discussion (Lines #384–387).

Reviewer #3 (Remarks to the Author):

General Comments: *Revised manuscript is suitable for publication.*

Response: Thank you very much for your endorsement of our manuscript!

REVIEWERS' COMMENTS

Reviewer #1 (Remarks to the Author):

The authors have fully addressed my last points.
I am looking forward to see this exciting work published.
Congratulations